# The transcription factor NF-Y participates to stem cell fate decision and regeneration in adult skeletal muscle

Giovanna Rigillo [1], Valentina Basile[1], Silvia Belluti [1], Mirko Ronzio[2], Elisabetta Sauta [3], Alessia Ciarrocchi[4], Lucia Latella[5], Marielle Saclier [2], Susanna Molinari[1], Antonio Vallarola[1], Graziella Messina [2], Roberto Mantovani[2], Diletta Dolfini [2] & Carol Imbriano [1✉]

The transcription factor NF-Y promotes cell proliferation and its activity often declines during differentiation through the regulation of NF-YA, the DNA binding subunit of the complex. In stem cell compartments, the shorter NF-YA splice variant is abundantly expressed and sustains their expansion. Here, we report that satellite cells, the stem cell population of adult skeletal muscle necessary for its growth and regeneration, express uniquely the longer NF-YA isoform, majorly associated with cell differentiation. Through the generation of a conditional knock out mouse model that selectively deletes the NF-YA gene in satellite cells, we demonstrate that NF-YA expression is fundamental to preserve the pool of muscle stem cells and ensures robust regenerative response to muscle injury. In vivo and ex vivo, satellite cells that survive to NF-YA loss exit the quiescence and are rapidly committed to early differentiation, despite delayed in the progression towards later states. In vitro results demonstrate that NF-YA-depleted muscle stem cells accumulate DNA damage and cannot properly differentiate. These data highlight a new scenario in stem cell biology for NF-Y activity, which is required for efficient myogenic differentiation.

---

[1] Department of Life Sciences, University of Modena and Reggio Emilia, via Campi 213/D, Modena, Italy. [2] Department of Biosciences, University of Milan, via Celoria 26, Milan, Italy. [3] Department of Electrical, Computer and Biomedical Engineering, University of Pavia, Pavia, Italy. [4] Laboratory of Translational Research, Azienda USL-IRCCS, Reggio Emilia, Italy. [5] Department of Medicine, Institute of Translational Pharmacology, Italian National Research Council and Epigenetics and Regenerative Medicine, IRCCS Fondazione Santa Lucia, Rome, Italy. ✉email: carol.imbriano@unimore.it

Postnatal muscle stem cells, namely satellite cells (SCs), allow neonatal/juvenile growth phase of normal skeletal myofibers as well as the homeostasis and repair following injury of adult skeletal muscle[1,2]. SCs are histologically located in a niche environment between the sarcolemma and the basal lamina of the muscle fiber. Despite heterogeneous in gene expression signature, stemness, and myogenic differentiation potential, SCs are characterized by the expression of the paired domain transcription factor Pax7[3]. The ablation of the total pool of Pax7+ cells in adulthood results in the complete loss of muscle regeneration[4,5]. In adult muscle, Pax7+ cells are mitotically quiescent but quickly enter the cell cycle in response to injury and muscle degeneration to both self-renew and differentiate into functional myofibers. A dynamic interplay exists between SCs and the stem cell niche: the activity of SCs is influenced not only by intrinsic factors, but also by signals provided by the niche[6]. In addition, the niche has a role in the regulation of SCs asymmetric division, which depends on the relative position of daughter cells in relation to the myofiber[7]. Asymmetric division, which generates two unequal daughter cells, is a key mechanism that allows the maintenance of the stem cells pool and tissue homeostasis by supporting the balance between self-renewal and differentiation[7].

Multiple signaling pathways and rapid expression of myogenic transcription factors (TFs) govern SCs activation. In particular, the early expression of Myf5 and MyoD in activated SCs controls the myogenic program, depending on which of the two TFs predominates: while translation of Myf5 mRNA enhances SCs activation, MyoD mainly drives the expansion of a cell population that is committed toward early differentiation[3,8]. After limited rounds of proliferation, the majority of Pax7+ cells begin a differentiation program characterized by MyoD-dependent activation of Myogenin and Mef2 and decrease in Pax7 levels[9].

The TF NF-Y is composed by NF-YA, NF-YB, and NF-YC subunits. NF-YA is the DNA binding subunit of the heterocomplex, which specifically binds to CCAAT boxes, common regulatory DNA elements within promoters and enhancers[10,11]. The NF-YA gene encodes for two different splice transcripts, NF-YA long (NF-YAl) and NF-YA short (NF-YAs), the last one lacking 28/29 aminoacids within the transactivation domain. In mouse embryonic stem cells (ESCs), NF-YAs is expressed in proliferating cells and a switch to NF-YAl occurs during differentiation[12].

In mature skeletal muscle, NF-YA expression is absent and, consequently, NF-Y binding to its target genes is lost[13]. NF-Y is a well-known positive regulator of cell proliferation via transcriptional activation of cell growth genes. Therefore, it is generally accepted that the decrease in NF-YA levels is a key step that allows the exit from the cell cycle and the activation of the myogenic differentiation program during myoblast differentiation. Indeed, transient NF-YA overexpression impairs myoblast differentiation in vitro[13,14]. Despite this, we recently showed that stable overexpression of NF-YAl in C2C12 myoblasts enhances the differentiation program through the activation of new identified NF-Y target genes, such as *Cdkn1c* (p57) and *Mef2d*, which are important for early muscle differentiation. Oppositely, forced expression of NF-YAs maintains the cells into a proliferative status and inhibits the differentiation program[15]. Recent results obtained by CRISPR-Cas9 forced deletion of exon3, retained in NF-YAl, highlighted that the switch from NF-YAl to NF-YAs interferes with the differentiation program but not with the proliferative ability of C2C12 cells[16]. While in wt mice muscle NF-YA is undetectable, it is expressed in the nuclei of dystrophic *mdx* mouse muscles, which are characterized by extensive activation of SCs to compensate myofibers degeneration. In accordance with NF-YA expression, the NF-Y heterotrimer binds the promoters of its target genes, which are upregulated in *mdx*

muscles[14]. Taking into consideration these results together with data showing that NF-YA expression controls stemness and proliferation of mouse and human stem cells[12,17], we decided to investigate the biology of NF-Y in muscle stem cells. Unfortunately, the ability to study NF-Y in post-natal tissues has been hampered by early embryonic lethality of NF-YA null mice[18].

In this study, we used an inducible knock out mouse model to selectively disrupt NF-YA expression in Pax7+ SCs and we demonstrated that NF-Y has key functions in the regulation of muscle stem cell fate. We present evidence that NF-Y is important for adult SCs maintenance and myogenic differentiation.

## Results

**NF-YA is required for proper regeneration of adult skeletal muscle following injury.** We first investigated the expression of the NF-YA subunit during embryonal and adult myogenesis. Western blot analysis of mouse skeletal muscles highlighted that NF-YA expression is high in embryonic myoblasts (E12) and drops in fetal (E17) and post-natal muscles from P7 to P28 (Fig. 1a). Despite NF-YA downregulation during secondary myogenesis and post-natal period, we decided to investigate whether NF-YA expression could be modulated during skeletal muscle regeneration induced in tibialis anterior (TA) of adult mice (6–8-weeks-old mice) by cardiotoxin (CTX) injection. CTX administration led to a significant increase in NF-YAl transcript and upregulation of both NF-YAl and NF-YAs protein isoforms in total muscle extracts (Fig. 1b and c). Immunofluorescence analysis of TA muscle sections after 5 days from CTX treatment identified NF-YA expression in some nuclei of centrally nucleated fibers, as well as in interstitial cells that populate regenerating muscles (Fig. 1d). The main populations involved in muscle regeneration were isolated from CTX muscles by cell sorting and total extracts were analyzed by western blot: while macrophages express both NF-YAl and NF-YAs proteins, NF-YAl isoform uniquely is detected in activated SCs and FAPs (Fibro Adipogenic Progenitors) (Fig. 1e).

In adult life, SCs are the stem cell population responsible for muscle regenerative capacity, therefore we decided to investigate whether NF-YA is required for the regeneration of myofibers upon injury. We generated NF-YA^fl/fl;Pax7-Cre mice by crossing Pax7-CreER^T2 [19] with NF-YA^flox/flox (NF-YA^fl/fl) mice, containing one LoxP site in intron 2 and one LoxP site in intron 8 of the NF-YA gene[18] (Supplementary Fig. 1a). Genetic ablation of NF-YA expression in SCs of NF-YA^fl/fl;Pax7-Cre mice was induced by five repeated intraperitoneal (IP) injections of Tamoxifen (TMX) followed by 7 days of chasing in 8 weeks old mice (hereafter NF-YA^cKO). NF-YA^fl/fl mice treated with TMX were used as control. PCR analysis of isolated SCs identified TMX-induced genomic targeted-deletion of the NF-YA floxed gene and a decrease in NF-YA wt transcript in favor of NF-YA deleted one (Supplementary Fig. 1b and c). Quantification of NF-YA and Pax7 double stained cells on fresh extensor digitorum longus (EDL) myofibers isolated from NF-YA^fl/fl and NF-YA^cKO mice showed TMX-induced depletion of NF-YA expression in about 70% of SCs (Supplementary Fig. 1d). NF-YA^cKO mice showed neither an overt phenotype nor differences in morphology and fiber size distribution compared to control (Supplementary Fig. 1e).

CTX was then injected into TA muscles of NF-YA^cKO and NF-YA^fl/fl mice. H&E staining highlighted a more severe muscle damage in NF-YA^cKO than in control mice at 5 days post-injection (Fig. 1f). At 15 days, centronuclear myofibers, indicative of regeneration, were detected in both NF-YA^fl/fl and NF-YA^cKO mice, but disorganized muscle structure and interstitial cell infiltration were clearly evident in cKO muscles. The reduction in

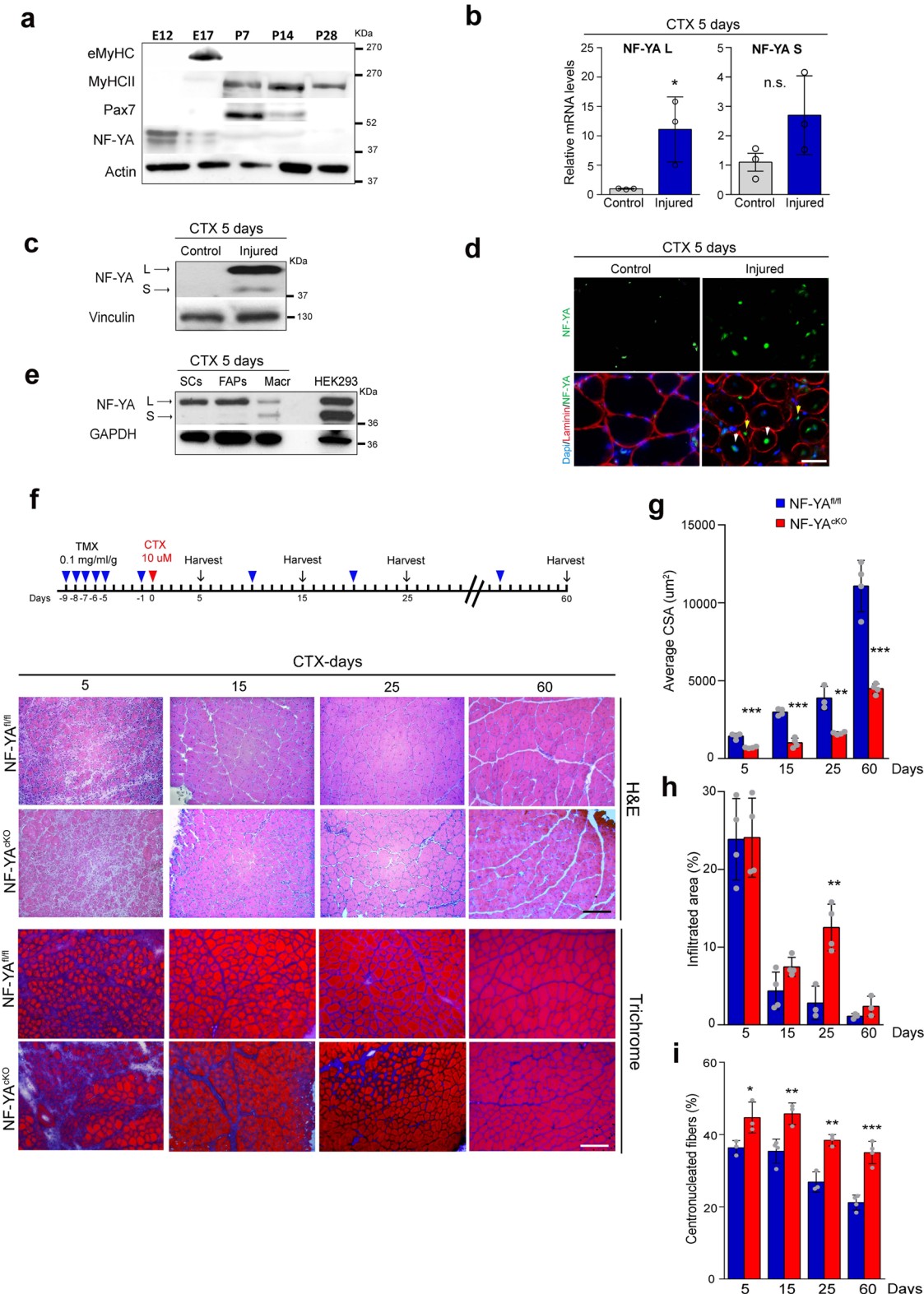

myofiber cross-sectional area (CSA) suggested a delayed regenerative response in NF-YA$^{cKO}$ mice (Fig. 1g). Moreover, cell infiltration persisted in NF-YA$^{cKO}$ sections (Fig. 1h), as well as collagen-enriched area, detected by trichrome staining, even at day 25 from injury (Fig. 1f). The difference in CSA and percentage of centrally nucleated myofibers between NF-YA$^{fl/fl}$ and NF-YA$^{cKO}$ mice was still evident following 60 days after injury and was associated with reduced muscle weight (Fig. 1g, I

and Supplementary Fig. 1f). Despite this, no defects were observed on muscle function, at least when determined by the conventional forelimb grip strength test (Supplementary Fig. 1g).

These results suggest that NF-YA expression in SCs could be essential for proper timing of skeletal muscle regeneration.

**NF-YA expression sustains proper regenerative myogenesis in vivo.** To investigate the effect of NF-YA decrease on

**Fig. 1 NF-YA is required for proper regeneration of adult skeletal muscle following injury. a** Western blot analysis of NF-YA protein levels in embryonic (E12), fetal (E17) and post-natal (from P7 to P28) skeletal muscles. MyHCII, eMyHC and Pax7 antibodies were used to follow the myogenic progression. $n = 3$ experiments. **b** Relative mRNA levels of NF-YA isoforms (L = long and S = short) in uninjured (control) and 5 days-injured whole TA muscle of WT mice measured by RT-qPCR. Data represent mean ± s.d. (two-tailed unpaired $t$-test: $p = 0.0343$; *$p < 0.05$; n.s. = not significant; $n = 3$ mice). **c** Representative immunoblot of NF-YA isoforms (arrows) in uninjured and 5 days-injured whole TA muscle extracts of WT mice. $n = 3$ experiments. **d** Immunofluorescence analysis of NF-YA and laminin in TA muscle cross-sections 5 days after CTX injury and uninjured control. White and yellow arrows indicate nuclei of regenerative myofibers and interstitial cells, respectively. $n = 3$ experiments, scale bar: 50 µm. **e** Western blot analysis of NF-YA protein expression in sorted satellite cells (SCs), fibro/adipogenic progenitors (FAPs) and macrophage (Macr) cells from CTX-injured TA muscle. Hek293 cell extract was used to mark the different molecular weights of NF-YA isoforms. $n = 3$ experiments. **f** Upper panel: Schematic representation of TMX and CTX administration and time-course sample collection. Lower panel: Representative images of H&E and trichrome-stained TA cross-sections of NF-YA^cKO compared with NF-YA^fl/fl mice at 5, 15, 25, and 60 days after CTX injury. $n = 3$ mice, scale bar: 100 µm. **g** Quantification of average cross-sectional area (CSA) (5 days $p = 0.0002$; 15 days $p < 0.0001$; 25 days $p = 0.0017$; 60 days $p = 0.002$) and **h, i** percentage of infiltrated area (25 days $p = 0.0052$) and centronucleated myofibers (5 days $p = 0.0380$; 15 days $p = 0.0082$; 25 days $p = 0.0034$; 60 days $p = 0.0003$) in TA muscle sections at days 5, 15, 25, and 60 after CTX injection. Data represent mean ± s.d. (two-tailed unpaired $t$-test was used to calculate the statistical significance: **$p < 0.01$, ***$p < 0.001$; $n = 4$ mice).

regenerative myogenesis, we evaluated the expression of key myogenic regulatory factors. Immunofluorescence staining of embryonic Myosin Heavy Chain (eMyHC), a marker for actively regenerating myofibers, showed marked expression at 5 days post-injury in both NF-YA^fl/fl and NF-YA^cKO muscles. NF-YA^fl/fl samples presented abundant expression of eMyHC in fibers of bigger size compared to NF-YA^cKO. We observed an aberrant persistence of numerous eMyHC+ myofibers at day 15 after injury in cKO-damaged muscles (Fig. 2a), evidence of a regenerative delay.

By means of RT-qPCR, we compared the transcriptional activation of myogenic markers following 5 days from injury in total muscles in both NF-YA^fl/fl and NF-YA^cKO mice, relatively to control uninjured muscles (Fig. 2b). All the examined transcripts were induced in NF-YA^fl/fl injured muscle, consistently with active regeneration. Differently, CTX administration to NF-YA^cKO muscles did not induce significant upregulation of Pax7 with respect to uninjured muscles. Myogenin, Myf5 and eMyHC levels increased, albeit to a lesser extent compared to NF-YA^fl/fl. Accordingly, reduced levels of Pax7, Myogenin and eMyHC proteins were observed in NF-YA^cKO compared to NF-YA^fl/fl CTX injured muscles (Fig. 2c). Western blot on 15 days post-injury muscles confirmed that the expression of eMyHC persisted at abundant levels in NF-YA^cKO with respect to NF-YA^fl/fl (Fig. 2d), consistently to the higher proportion of eMyHC+ fibers observed by immunofluorescence (Fig. 2a).

These results are suggestive of a delayed regenerative process in NF-YA^cKO mice, hinting that NF-YA expression is important for efficient muscle repair in adult mice.

**NF-YA expression is regulated during SCs activation and differentiation.** To understand how NF-Y regulates post-natal myogenesis and muscle homeostasis, we examined the expression of NF-YA in SCs retained on single isolated myofibers from EDL uninjured muscles of adult wt mice (Fig. 3a). In order to discriminate quiescent, proliferating, and differentiating SCs we fixed myofibers after isolation (quiescent, Q), after 1 day (proliferating, P), and 3 days in suspension culture (differentiating, D). We detected positive NF-YA staining in quiescent Pax7+, activated MyoD+, and differentiating MyoG+ SCs. The analysis of differentiating fibers stained with Pax7 highlighted that NF-YA is expressed also in self-renewing (SR) SCs[20] (Fig. 3a).

We investigated the levels of NF-YA transcripts in SCs enriched from dissociated skeletal muscle of wt mice cultured in vitro. As already observed in immortalized myoblasts[15], NF-YAl is the most abundant transcript expressed in SC-derived primary myoblasts (Fig. 3b). Western blot analysis of total cellular extracts from proliferating SCs exclusively detected NF-YAl

protein (Fig. 3c). The expression of NF-YAl declines following 1 day of differentiation, concomitantly to Pax7 and MyoD decrease and Myogenin increase, whose modulation occurs at the transcriptional level (Fig. 3d). NF-YAl protein was barely detectable after 5 days of differentiation, when MyHC was highly expressed (Fig. 3c). The transcriptional control of NF-YA expression was validated also in SCs isolated from diaphragms of adult Pax3 GFP/+ mice[21] by flow cytometry and RNA extracted immediately after their sorting or following in vitro culturing. Relative quantification of mRNA levels confirmed a severe decrease of NF-YA transcripts during differentiation (Supplementary Fig. 2a).

**Disruption of NF-YA expression reduces SCs pool.** Using the NF-YA^cKO mouse model, we examined whether NF-YA participates to SCs fate decision in uninjured muscle. Immunofluorescence analysis revealed a clear decrease in the number of Pax7+ cells in TA cross-sections of NF-YA^cKO with respect to control mouse muscles (Fig. 4a). The comparison between the number of FACS-isolated SCs from control and cKO muscles corroborated that impaired NF-YA expression triggers SCs depletion (Fig. 4b). Tunel assay and cleaved-Caspase3 immunofluorescence staining showed a significant increase in Pax7+ myonuclei of NF-YA^cKO compared to NF-YA^fl/fl muscles, hinting that the reduction of Pax7+ cells can be at least in part ascribed to cell death (Fig. 4c and d).

Relative percentages within NF-YA^cKO myogenic cells in uninjured TA sections highlighted a significant decrease of quiescent Pax7+/MyoD- population (from 94% in NF-YA^fl/fl to 90% in NF-YA^cKO) at the expense of activated Pax7+/MyoD+ one (from 6% to 10%) (Fig. 4e). Consistently, we observed a higher number of Pax7+/EdU+ cells in NF-YA^cKO TA sections following in vivo EdU administration (from 1.75% ± 0.39 to 9.61% ± 1.29) (Fig. 4f). These data suggested that NF-YA^cKO SCs are prone to exit the quiescence. Similar results were obtained in CTX-injured muscles, where Pax7+/EdU+ cells increased from about 33% in control muscles to 54% in NF-YA^cKO ones (Supplementary Fig. 1h).

To better understand how NF-YA loss affects SCs fate, we monitored their myogenic progression in isolated EDL myofibers. Similar to what was observed in TA sections, the frequency of Pax7+ cells on total nuclei was clearly reduced in NF-YA^cKO myofibers at day 0 (d0), compared to NF-YA^fl/fl ones (Fig. 4g). Despite not statistically significant, relative cell percentages at d0 showed a decrease in Pax7+/MyoD- in favor of Pax7+/MyoD+ population, suggesting that the isolation procedure can activate NF-YA^cKO SCs from quiescence more quickly than NF-YA^fl/fl cells (Fig. 4h). After one day in culture (d1), NF-YA^cKO

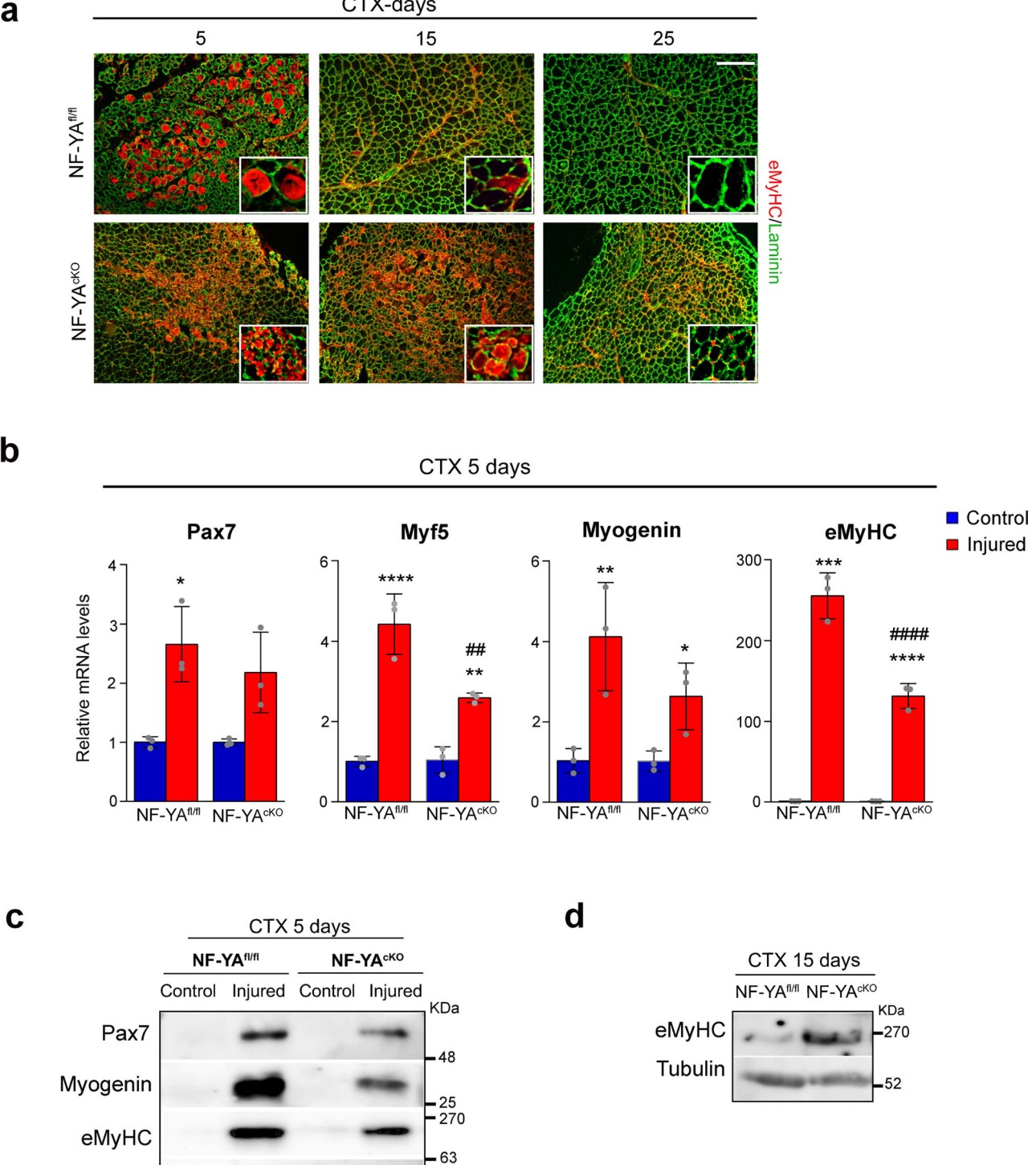

**Fig. 2 Loss of NF-YA delays regenerative myogenesis in vivo. a** Representative immunostaining for eMyHC (red) and laminin (green) on TA muscle sections of NF-YA^fl/fl and NF-YA^cKO mice 5, 15 and 25 days after CTX injury. Nuclei were identified by DAPI staining. Scale bar: 100 μm. **b** Relative mRNA levels of myogenic markers Pax7, Myf5, Myogenin, eMyHC in whole 5-days injured TA muscles of NF-YA^fl/fl and NF-YA^cKO mice measured by RT-qPCR. Transcript levels of NF-YA^fl/fl and NF-YA^cKO uninjured mice have been arbitrarily set at 1. Data represent mean ± s.d. (one-Way Anova: Pax7 $F_{(3,8)}$=9.66, $p = 0.0049$; Myf5 $F_{(3,8)}$=44.63, $p < 0.0001$; MyoG $F_{(3,8)}$=9,99, $p = 0.0044$; eMyHC $F_{(3,8)}$=170.8, $p < 0.0001$; $n = 3$ mice per group. $*p < 0.05$, $**p < 0.01$, $***p < 0.001$, $****p < 0.0001$ vs control; $^{\#\#}p < 0.01$, $^{\#\#\#\#}p < 0.0001$ vs NF-YA^fl/fl). **c** Western blot analysis performed on whole protein extracts of uninjured (control) and 5-days injured TA muscles of NF-YA^fl/fl and NF-YA^cKO mice. Immunoblots represent protein levels of Pax7, eMyHC, Myogenin and Tubulin. $n = 3$ experiments. **d** Western blot analysis of eMyHC on whole protein extracts of 15-days injured TA muscles of NF-YA^fl/fl and NF-YA^cKO. Tubulin was used as loading control. $n = 3$ experiments.

**a**

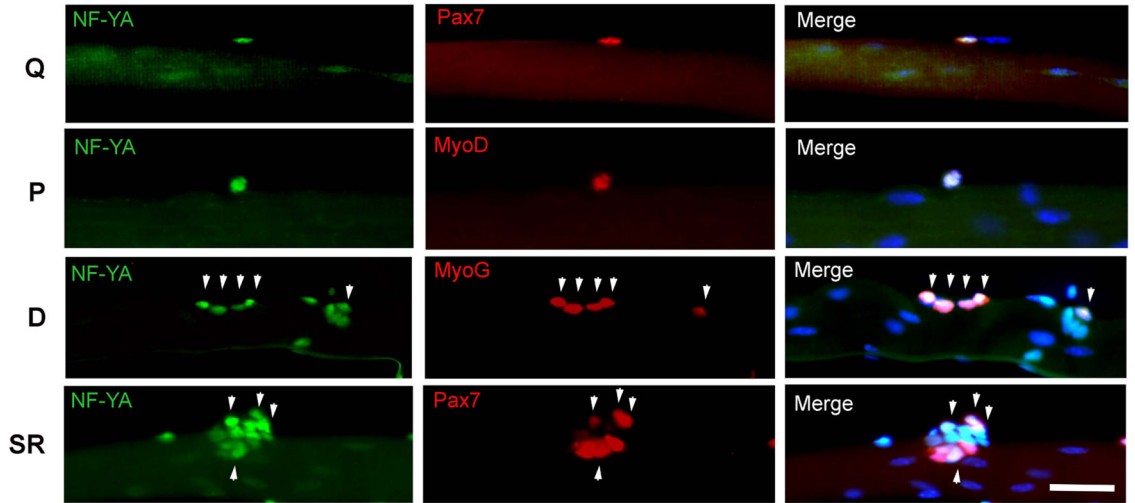

**b**

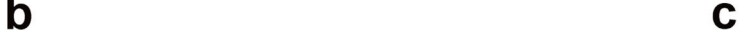

**c**

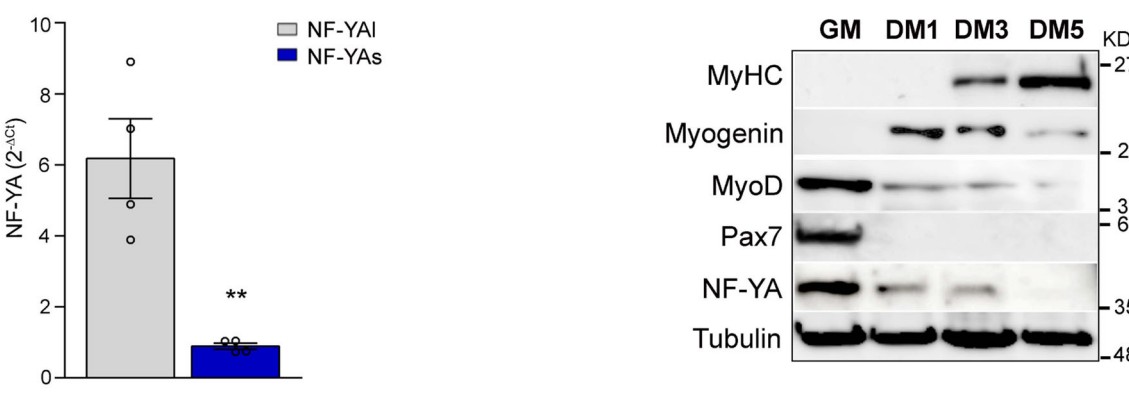

**d**

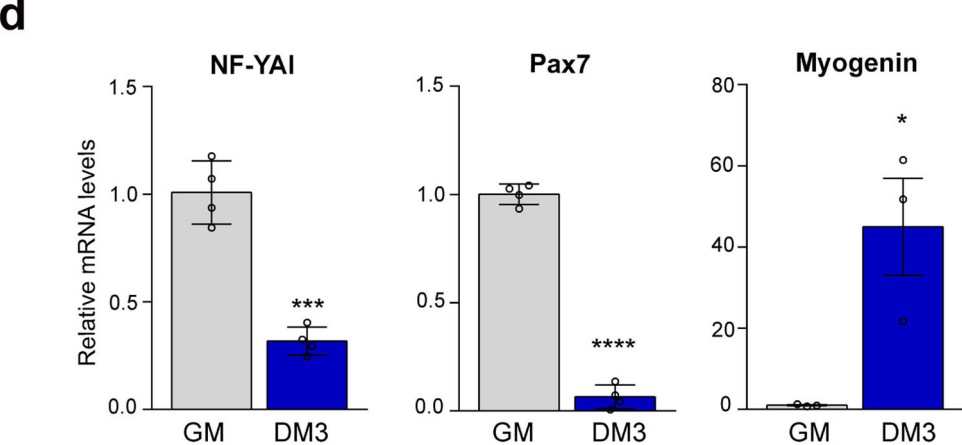

fibers showed a significant increase in Pax7−/MyoD+ cells (62.3%) compared to NF-YA$^{fl/fl}$ control cells (53.4%), suggesting a quick transit of SCs from proliferation towards differentiation commitment. At day 2 (d2), NF-YA$^{fl/fl}$ cells were equally distributed between MyoD+/MyoG− and MyoD+/MyoG+ populations, while the balance was shifted towards MyoD+/MyoG+ (65%) in NF-YA$^{cKO}$ fibers. At day 3 (d3), the distribution of NF-YA$^{cKO}$ cells was similar to the previous time point, with 67% of the population still stained for MyoD+/MyoG+ and about 30% MyoD+/MyoG−. Only 3% of the MyoG+ cells switched off the expression of MyoD (MyoD−/MyoG+). Differently, MyoD+/MyoG+ cells continued to

**Fig. 3 NF-YA expression is regulated during satellite cells activation and differentiation. a** NF-YA immunofluorescence in Pax7+ SCs on freshly isolated EDL myofibers (quiescent, Q), MyoD-activated SCs after 1 day in suspension culture (proliferating, P), MyoG+ differentiating (D) and Pax7+ self-renewing (SR) SCs after 3 days in suspension culture. Nuclei were identified by DAPI staining. n = 3 mice, scale bar: 50 μm. **b** mRNA levels of NF-YAl and NF-YAs transcripts in isolated SCs measured by RT-qPCR. Data represent mean ± s.d. (two-tailed unpaired *t*-test: p = 0.0033, *p < 0.05, n = 4 mice). **c** Protein expression analysis by Western blot of NF-YA and myogenic markers (Pax7, MyoD, MyoG and MyHC) in SC-enriched cultures maintained in proliferating (GM) or differentiating conditions for 1, 3 or 5 days (DM1, DM3, DM5). Tubulin was used as loading control. n = 3 experiments. **d** Relative mRNA levels of NF-YAl (p = 0.0001, n = 4 mice), Pax7 (p < 0.0001, n = 4 mice) and Myogenin (p = 0.0212, n = 3 mice) in proliferating (GM) and differentiating (DM3) SCs cultured in vitro. Transcript levels in GM condition have been arbitrarily set at 1. Data represent mean ± s.d. (two-tailed unpaired *t*-test: *p < 0.05, ***p < 0.001, ****p < 0.0001).

increase (from 48 to 67%) at the expense of MyoD+/MyoG− activated population (from 51.2 to 18.8%) in NF-YA[fl/fl] myofibers. Moreover, the appearance of 14.3% MyoD−/MyoG+ cells indicated their progression throughout the differentiation program.

As disruption of NF-Y activity resulted in the loss of cellular quiescence and quick commitment towards differentiation, we assessed whether some quiescent cells executed a G0-to-differentiation transition without passing through S phase. EdU pulse-chase experiment on differentiating myofibers (d3) did not identify MyoG+/EdU− cells, definitely confirming that DNA replication occurred also in NF-YA[cKO] cells before they undertook differentiation (Fig. 4i). Despite not significant, the higher percentage of MyoG− cells within EdU+ population in NF-YA[cKO] fibers is consistent with the accumulation of cells still proliferating (Fig. 4h).

**Loss of NF-Y activity impairs SCs differentiation in vitro.** We decided to monitor the proliferation and differentiation abilities of SCs isolated from skeletal muscle of NF-YA[cKO] mice and cultured in vitro. Equal number of isolated cells from NF-YA[cKO] and NF-YA[fl/fl] muscles were plated and grown in proliferating conditions. Cultures derived from NF-YA[cKO] contained a significant lower proportion of Pax7+ cells when quantified on total DAPI-stained nuclei (Fig. 5a, b). Relative quantification within Pax7+ cells showed an increase in proliferating Pax7+/MyoD+ SCs (from 55.4% in control to 70.4% in NF-YA[cKO]) at the expense of quiescent Pax7+/MyoD− cells in NF-YA[cKO] (from 44.6 to 29.6%) (Fig. 5b).

These data suggested that, despite reduced in percentage, Pax7+ cell population from NF-YA[cKO] is more prone to activation (Pax7+/MyoD+). Consistently, we observed an increase in EdU+/Pax7+ cells, hinting that primary myoblasts from NF-YA[cKO] muscle are stimulated to proliferate when cultured in growth medium (Fig. 5c).

Despite the presence of proliferative markers, NF-YA[cKO] SCs underwent a modest expansion compared to control cells, even if maintained in culture for more days. This suggested that following DNA replication, cell division did not really occur. To further analyze the impact of NF-YA disruption on SCs proliferation, we performed cell cycle analysis by flow cytometry of Propidium Iodide-stained cells cultured in growth medium. We observed a significant increase in S-phase cells and the accumulation in G2/M at the expense of G0/G1 population (Fig. 5d). The expression of the mitotic marker phospho(Ser10)-H3 did not change compared to control cells, hinting that NF-YA[cKO] SCs could be arrested in S or G2 phase (Fig. 5e).

We then investigated the differentiation commitment of SCs cultured in low serum condition following 5 days from isolation: fewer MyHC+ were observed in NF-YA[cKO] culture compared to NF-YA-expressing cells (Fig. 5f). In order to achieve the same confluence, NF-YA[cKO] cells were maintained in culture for more days or were plated at higher density before switching to the differentiation medium: despite this, SCs failed to efficiently

differentiate (Fig. 5f, g). Furthermore, we stained myogenic cells with MyoD and MyoG antibodies after 1 or 2 days in differentiating conditions (DM1 or DM2). At DM2, NF-YA[cKO] cells showed different distribution compared to NF-YA[fl/fl] ones (Fig. 5h). MyoD+/MyoG+ committed population increased from 49.5% ± 1.2 in NF-YA[fl/fl] to 63.5% ± 2.1 in NF-YA[cKO] at the expense of more differentiated MyoD−/MyoG+ cells (from 32.7% ± 2.2 in NF-YA[fl/fl] to 18.2% ± 2.6 in NF-YA[cKO]). Western blot analysis of total cellular extracts showed reduced expression of Pax7 and MyHC in NF-YA[cKO] SCs cultured in growth or differentiating medium for three days (DM3) (Fig. 5I).

These results demonstrate that NF-YA[cKO] SCs experienced defects in the differentiation progression in vitro, as observed in isolated myofibers (Fig. 4h).

**Identification of NF-Y-regulated transcriptome in SCs.** To better understand the phenotype observed in primary NF-YA[cKO] myoblasts, we performed RNA-seq analysis of SCs isolated from NF-YA[fl/fl] and NF-YA[cKO] mice and cultured in growth medium. We defined differentially expressed genes with stringent criteria (see "Methods") and we performed a functional analysis of downregulated and upregulated genes. Gene Ontology analysis identified cell cycle-related pathways as the most significantly upregulated terms in cKO cells, in opposition to muscle cell differentiation and contraction categories that were evidently downregulated compared to control cells (Fig. 6a). The analysis of TFBS (Transcription Factor Binding Sites) over-represented in promoters (−450/+50) of differentially expressed genes identified NF-YA and NF-YB as enriched binding sites in upregulated genes but not in downregulated ones (Fig. 6b). These results are consistent with the hypothesis that NF-Y does not directly regulate the expression of genes involved in muscle differentiation[16], while it directly controls cell cycle genes.

We validated RNA-seq data by RT-qPCRs analysis (Supplementary Fig. 3a). In accordance with RNA-seq, CcnB2, CcnA2 and TopoIIα genes, encoding for key regulators of the cell cycle, were upregulated in NF-YA[cKO] primary myoblasts. In opposition, the myogenic genes Mef2C, MyoD, and Myogenin were downregulated. The transcription of the cell cycle inhibitors (CDKIs) Cdkn1A (p21), Cdkn2A (p16), and Cdkn1C (p57), which are important in the control of proper myogenic progression, decreased in cKO versus control cells. Taking into consideration the reduction of Pax7+/MyoD− SCs in uninjured NF-YA[cKO] muscles (Fig. 4e), we further investigated the mRNA levels of genes implicated in quiescence and self-renewal processes (Supplementary Fig. 3a). The expression of Sprouty1 (Spry1), which is required for SCs self-renewal and is downregulated during proliferation, significantly decreased in NF-YA[cKO] compared to NF-YA[fl/fl] primary myoblasts. Similarly, transcript levels of Notch1 and HeyL, whose knock out decreases the number of quiescent SCs in postnatal adult muscle[22,23], dropped in NF-YA[cKO]. We also analyzed the expression of NF-Y subunits: despite not significant, NF-YA decrease was concomitant to a transcriptional upregulation of the NF-YB and NF-YC genes, as

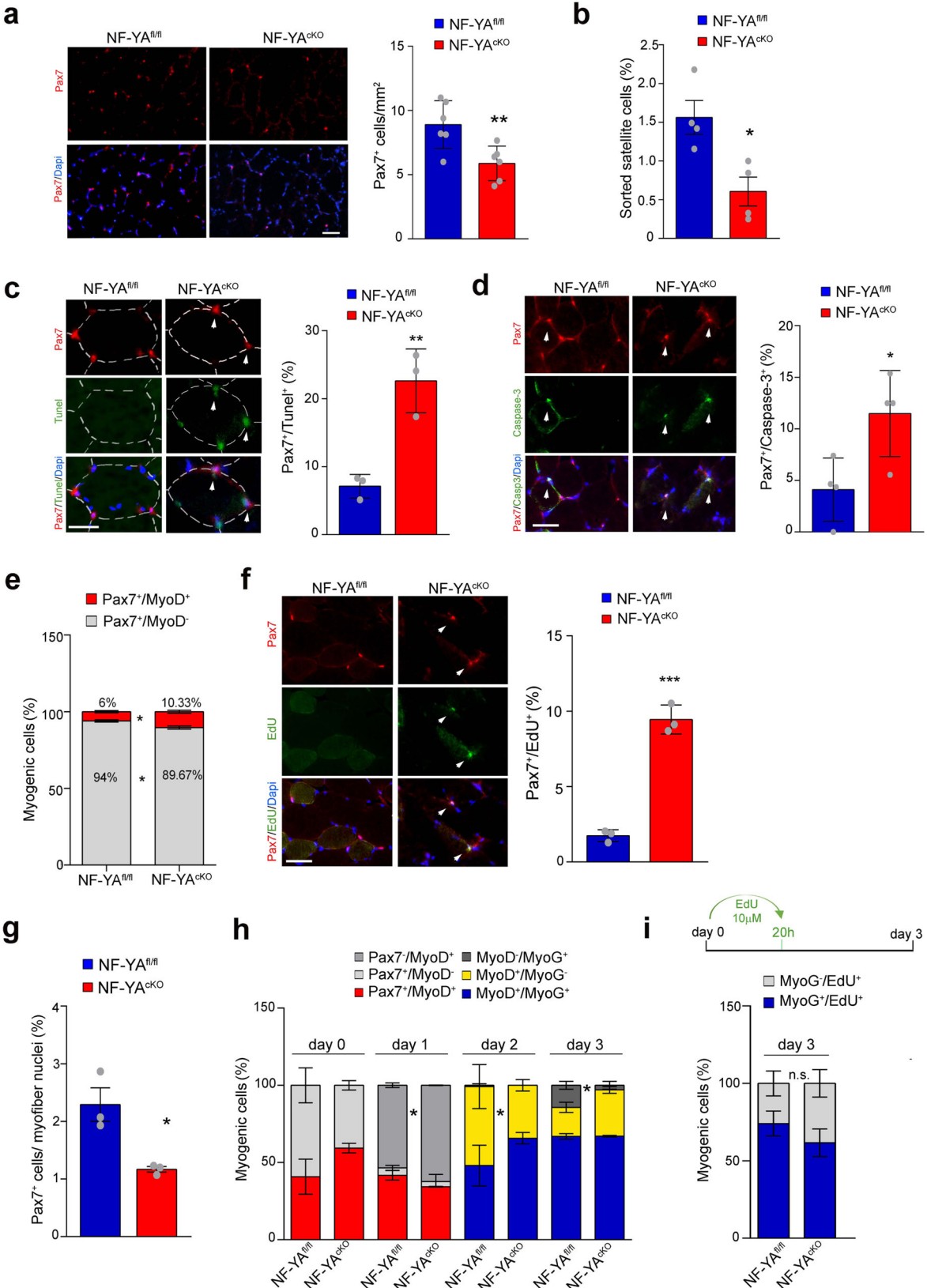

shown in other conditional NF-YA knock-out mouse models because of the negative transcriptional feedback existing between NF-Y subunits[24–26] (Supplementary Fig. 3b). The effects of NF-YA KO on the expression levels of p57, Myogenin and NF-YB were confirmed by western blot on total SCs extracts (Supplementary Fig. 3c).

Since the upregulation of CCAAT-genes of cell cycle categories in NF-YAcKO cells was unexpected, we decided to determine whether NF-Y was still recruited on their regulatory regions. We performed Chromatin IPs (ChIPs) on SCs cultured in the same conditions used for RNA-seq profiling. The binding of NF-YA to promoters of CcnB1, CcnB2 and TopoIIα genes was significantly

**Fig. 4 NF-YA loss reduces satellite cells pool and affects myogenic progression. a** Pax7 immunostaining performed on NF-YA$^{fl/fl}$ and NF-YA$^{cKO}$ TA muscle cross-sections (left panel) and related quantification per mm$^2$ (right panel). Nuclei were identified by DAPI staining. Scale bar: 50 μm. The histogram represents mean ± s.d. (two-tailed unpaired $t$-test: $p = 0.0091$, $*p < 0.05$; $n = 6$ mice). **b** Satellite cells were isolated from digested hindlimb muscles of NF-YA$^{fl/fl}$ and NF-YA$^{cKO}$ mice and sorted after incubation with anti-SM/C-2.6 and anti-CD45. SCs positive to SM/C-2.6 and negative to CD45 antigens were normalized on total sorted cells and expressed in percentage. Data represent mean ± s.d. (two-tailed unpaired $t$-test: $p = 0.0158$ $*p < 0.05$, $n = 4$ experiments). **c** Left panel: representative images of Tunel (green) and Pax7 (red) positive myonuclei in TA cross sections from NF-YA$^{fl/fl}$ and NF-YA$^{cKO}$ mice. Nuclei were identified by DAPI staining. The arrows indicate double stained cells. Scale bar: 25 μm. Right panel: percentage of Tunel/Pax7 double-positive cells. Data represent mean ± s.d. (two-tailed unpaired $t$-test: $p = 0.005$, $**p < 0.01$; $n = 3$ mice). **d** Representative images (left panel) and quantification (right panel) of Pax7/cleaved-Caspase-3 double stained cells in NF-YA$^{fl/fl}$ and NF-YA$^{cKO}$ TA cross sections. Scale bar: 50 μm. Data represent mean ± s.d. (unpaired $t$-test: $p = 0.029$, $*p < 0.05$, $n = 4$ mice). **e** Relative quantification of myogenic cells in TA cross sections from NF-YA$^{fl/fl}$ and NF-YA$^{cKO}$ uninjured mice. Data represent mean ± s.d. (two-tailed unpaired $t$-test: $p = 0.0147$; $*p < 0.05$; $n = 3$ mice). **f** Left panel: representative images of EdU (green) and Pax7 (red) positive myonuclei in TA cross sections from NF-YA$^{fl/fl}$ and NF-YA$^{cKO}$ uninjured mice. Scale bar: 50 μm. EdU (5 mg/Kg) was injected 24 h before sacrifice. Right panel: percentage of EdU + /Pax7+ cells in uninjured muscles. Data represent mean ± s.d. (two-tailed unpaired $t$-test: $p = 0.0013$, $**p < 0.01$; $n = 3$ mice). **g** Quantification of the number of Pax7+ SC nuclei per 100 myofiber nuclei in NF-YA$^{fl/fl}$ and NF-YA$^{cKO}$ muscles. Data represent mean ± s.d. (two-tailed unpaired $t$-test: $p = 0.0190$, $*p < 0.05$; $n = 3$ mice). **h** Relative distribution of myogenic cells retained in cultured EDL myofibers at day 0, 1, 2 and 3 post- isolation. Data represent mean ± s.d. (two-tailed unpaired $t$-test: $p$ values from left to right: $p = 0.0142$, $p = 0.0161$, $p = 0.0327$; $*p < 0.05$; $n = 3$ mice). **i** Relative quantification of myogenic cells after incubation for 20 h with EdU (10 μM), cultured for 3 days from myofibers isolation. Data represent mean ± s.d. (two-tailed unpaired $t$-test: n.s. = not significant; $n = 3$ mice).

reduced in NF-YA$^{cKO}$ compared to control SCs (Fig. 6c). These results hint that transcriptional activators other than NF-Y could be involved in maintaining active gene expression of cell cycle genes in muscle SCs. Similarly, we checked for NF-Y binding to regulatory regions of *Cdkn1C*, which we previously proved to be a NF-Y target gene[15]. In NF-YA$^{cKO}$ SCs, we did not observe a decrease in NF-YA binding to the canonical promoter but to the CCAAT-box within the 5′UTR, which has a demonstrated transcriptional regulatory function[27] (Fig. 6d). Moreover, since Myogenin is a critical node that drives myogenesis, we better characterized its regulatory regions. We previously demonstrated that *Myogenin* has a CCAAT-less promoter that coherently is not bound by NF-Y[15]. Three enhancer elements controlling *Myogenin* transcription have been described, E1, E2, and E3, located up to 7 kb from the TSS (Transcription Start Site)[28,29], with E3 uniquely containing a NF-Y binding site (Supplementary Fig. 3d). ChIP results identified NF-YA binding to the E3 enhancer that significantly decreased in NF-YA$^{cKO}$ cells (Fig. 6e). Among genes directly involved in stem cells quiescence, we analyzed whether NF-Y directly controls Sprouty1, because of its pivotal role in the maintenance of quiescence both in uninjured muscle and in the return to quiescence of SCs during muscle regeneration. NF-Y binding to the inverse CCAAT-box located at about 200 bp from the TSS (Supplementary Fig. 3e) significantly decreased in NF-YA$^{cKO}$ SCs (Fig. 6f).

Overall, the transcriptional profile of NF-YA$^{cKO}$ cells corroborated their impairment in the differentiation program through transcriptional modulation of indirect or direct target genes, such as *Cdkn1C* and *Myogenin*.

**NF-YA depletion induces DNA damage in SCs**. The analysis of larger pathway gene sets (>500 genes) highlighted the activation of GO pathways related to cell death/apoptosis and DNA damage/integrity in NF-YA$^{cKO}$ *versus* NF-YA$^{fl/fl}$ muscle cells (Fig. 7a). In particular, G2, intra-S, and mitotic-G2 DNA damage checkpoint signatures were retrieved (Fig. 7b). By RT-qPCRs we validated deregulated transcription levels of selected genes, such as *Mre11a* and *Rad51*, involved in homologues recombination (HR) generally restricted to the S and G2 phases, *Brca1* that belongs to the DNA damage response process (DDR), *Birc5* and *Bcl2l2* that participate to the apoptotic pathway (Fig. 7c).

Taking into consideration that NF-YA loss induces DNA damage in other cell types[30] and DNA damage affects the differentiation process[31,32], we investigated whether the accumulation of DNA damage can be one of the causes of impaired

differentiation in NF-YA$^{cKO}$ cells. We used S139-phosphorylated histone H2AX (γH2AX) as a marker of double-strand breaks (DSBs)[33]. In growth conditions, we observed a higher number of γH2AX+/Pax7+ cells in NF-YA$^{cKO}$ cultures (Fig. 7d). Moreover, western blot analysis of total extracts from NF-YA$^{cKO}$ compared to NF-YA$^{fl/fl}$ SCs corroborated an increase in the expression of γH2AX, as well as of MRE11A and RAD51 (Fig. 7e). Interestingly, the transduction of NF-YA$^{cKO}$ cells with NF-YAl-lentiviral particles decreased *Mre11a* and *Rad51* transcriptional activation (Fig. 7f), which reflects on protein levels (Fig. 7g). We concomitantly observed γH2AX protein to be reduced as well (Fig. 7g).

Therefore, our data demonstrate that NF-YA loss triggers DNA damage in proliferating adult muscle stem cells.

## Discussion

The activity of NF-Y has been extensively associated to cell cycle progression and proliferation through the binding and direct transcriptional activation of target CCAAT-containing genes. Thus, it is not surprising that NF-YA, the DNA binding subunit of the complex, is highly expressed in proliferating cells, among which stem cells, and declines during differentiation. The CCAAT box is one of the most overrepresented cis-regulatory elements in human and mouse ESCs and NF-Y is necessary for their survival and proliferation[12,34,35]. Jothi's work recently highlighted that NF-YA depletion in mouse ESCs leads to altered structural architecture and correct positioning of the transcriptional machinery at CCAAT-promoters and hence gives rise to ectopic transcripts[36].

Also in the skeletal muscle, active NF-Y ensures myoblasts proliferation and it then becomes inactive during differentiation through NF-YA downregulation[13]. In neonatal cardiomyocytes, characterized by regenerative ability after injury, NF-YA has been recently identified as the TF that, together with NFE2L1, efficiently induces cell proliferation and survival[37].

Another mechanism controlling NF-Y activity during proliferation/differentiation processes relies on NF-YA splice variants, NF-YAs and NF-YAl. In human hematopoietic stem cells and mouse ESCs, the major NF-YA isoform expressed in proliferating conditions is the shorter one, therefore NF-YAs has been pointed out as the stem splice variant[12,17,38]. A switch from NF-YAs to NF-YAl occurs during differentiation of ESCs and the expression of NF-YAs in immature c-kit+ bone marrow cells declines rapidly with terminal myeloid differentiation[12,17,38].

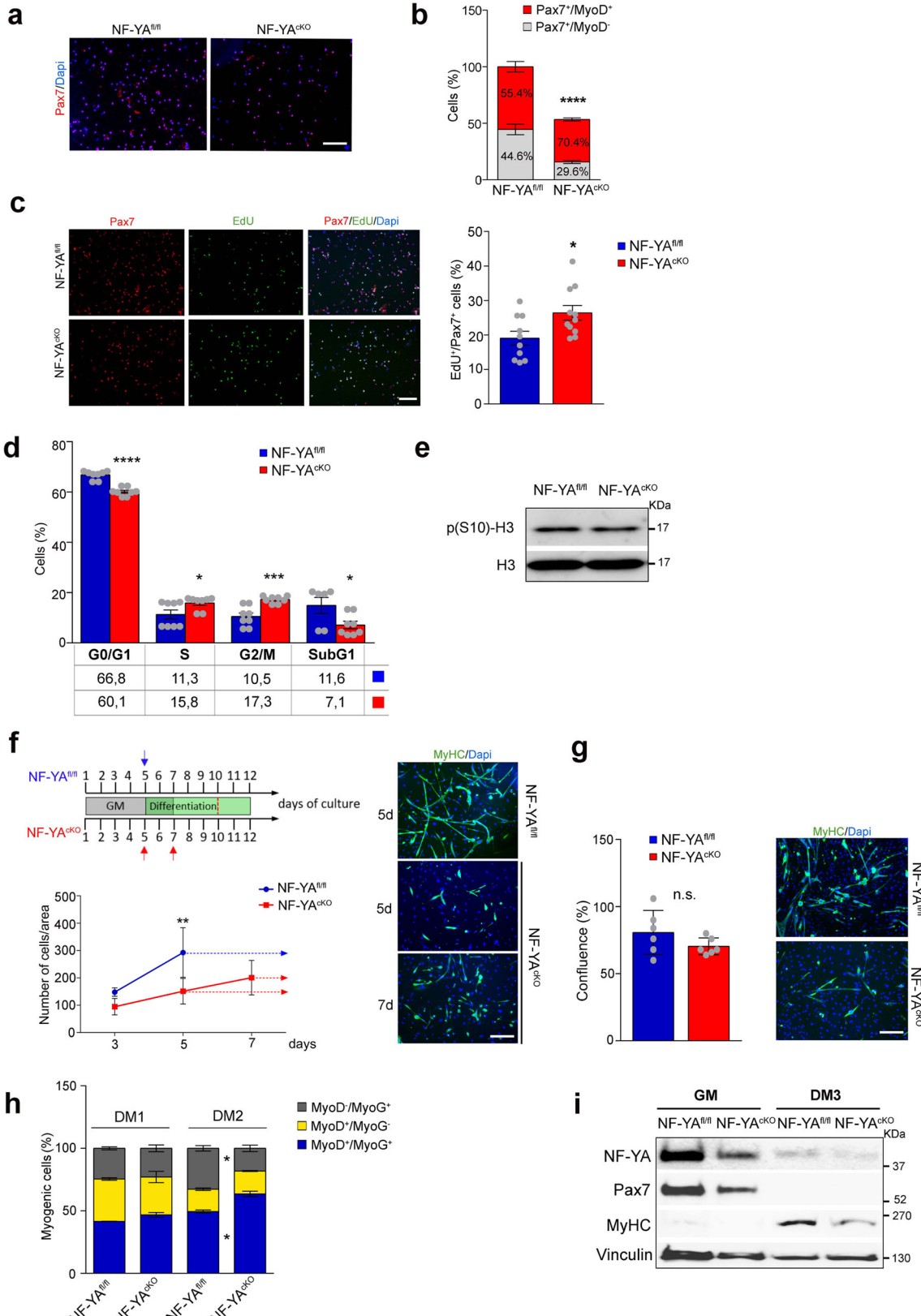

All together, these data suggest that NF-Y activity is high in mitotic cells and declines during differentiation through NF-YA regulation. Despite this, NF-Y expression and activity are critical for the homeostasis of post-mitotic cells, such as mature neurons of adult mouse brain[24] and postnatal differentiated mouse hepatocytes[25] or adipocytes[39].

Our results indicate that, in adult skeletal muscle, NF-YA is expressed by SCs but not in the nuclei of mature myofibers. Differently from what expected, muscle SCs express NF-YAl and not NF-YAs, as generally observed in stem cells[12,17]. The dynamics of NF-YA expression, high in proliferating and low in differentiating SCs, suggested that NF-Y activity could be

**Fig. 5 Loss of NF-Y activity impairs satellite cells differentiation in vitro. a** Representative images of NF-YA$^{fl/fl}$ and NF-YA$^{cKO}$ primary SCs immunostained for Pax7 and DAPI after 4 days from isolation ($n = 3$ experiments). Scale bar: 100 µm. **b** Quantification of the percentage of Pax7 positive cells normalized on total DAPI-stained nuclei in proliferating culture conditions. The indicated percentage is related to the distribution of Pax7+/MyoD+ and Pax7+/MyoD− populations. Data represent mean ± s.d. (two-tailed unpaired $t$-test: ****$p < 0.0001$, $n = 3$ mice). **c** Immunostaining (left panel) and related quantification (right panel) of EdU+/Pax7+ cells in proliferating SCs cultures isolated from NF-YA$^{fl/fl}$ and NF-YA$^{cKO}$. EdU (10 µM) was administered to the cells for 2 h until harvest. Data represent mean ± s.d. (two-tailed unpaired $t$-test: $p = 0.0215$; *$p < 0.05$, $n = 10$ mice). Scale bar: 100 µm. **d** Flow cytometry analysis of cell cycle progression in NF-YA$^{fl/fl}$ and NF-YA$^{cKO}$ SCs after 4 days from isolation. The histogram shows the distribution of cells in G0/G1 ($p < 0.0001$), S ($p = 0.0392$), G2/M ($p = 0.0005$) and subG1 ($p = 0.0336$) phases of the cell cycle. Data represent mean ± s.d. (two-tailed unpaired $t$-test: *$p < 0.05$, ***$p < 0.001$, ****$p < 0.0001$, $n = 8$ mice). **e** Western blot analysis of phospho-Ser10 histone H3 protein levels in whole-cell extracts of proliferating SCs. Total H3 was used as loading control. $n = 4$ experiments. **f** Upper panel: schematic representation of cell culture conditions and incubation times. GM = growth medium. Lower panel: proliferation curve at the indicated time points of NF-YA$^{fl/fl}$ and NF-YA$^{cKO}$ SCs maintained in growth medium (solid line) or in differentiating medium (dashed lines). Data represent mean ± s.d. (two-tailed unpaired $t$-test: day 5 $p = 0.0041$; **$p < 0.05$; $n = 3$ mice). Right panel: representative immunofluorescence images of MyHC staining of NF-YA$^{fl/fl}$ and NF-YA$^{cKO}$ SCs induced to differentiate for 5 days following 5 or 7 days from isolation. $n = 3$ mice, scale bar: 100 µm. **g** Left panel: percentage of confluence achieved by plating different NF-YA$^{fl/fl}$ and NF-YA$^{cKO}$ cell number. Data represent mean ± s.d. (two-tailed unpaired $t$-test: n.s. = not significant; $n = 3$ mice). Right panel: MyHC staining of NF-YA$^{fl/fl}$ and NF-YA$^{cKO}$ confluent SCs. **h** Relative distribution of NF-YA$^{fl/fl}$ and NF-YA$^{cKO}$ myogenic cells induced to differentiate for 1 (DM1) or 2 days (DM2) and stained with MyoD and MyoG antibodies. Data represent mean ± s.d. (two-tailed unpaired $t$-test: DM2 MyoD−/MyoG+ $p = 0.0423$, DM2 MyoD+/MyoG+ $p = 0.0255$; *$p < 0.05$; $n = 3$ mice). **i** Representative immunoblots of NF-YA and myogenic markers Pax7 and MyHCII protein expression in whole extracts of SCs isolated from NF-YA$^{fl/fl}$ and NF-YA$^{cKO}$ mice and cultured in growth medium (GM) and 3 days in differentiating medium (DM3). Vinculin was used as loading control. $n = 3$ experiments.

required for stem cell proliferation rather than differentiation. Despite this prediction based on NF-YA expression profile, here we described how NF-YA-depleted SCs are impaired in the differentiation process.

Through the generation of a conditional NF-YA$^{cKO}$ mouse model, in which NF-Y loss of function is exclusively induced in Pax7+ muscle cells, we investigated the fate of SCs in uninjured and injured muscle conditions. Uninjured muscles from NF-YA$^{cKO}$ mice experience a decline in SCs pool (~40%), caused at least in part by the activation of apoptotic cell death. Time course analysis of isolated myofibers allowed us to better understand naturally occurring activation and differentiation of SCs when retained within their microenvironment. NF-YA$^{cKO}$ cells are spontaneously activated and are quickly committed to early differentiation state. However, they are delayed in subsequent steps, as demonstrated by the lower number of MyoD−/MyoG+ cells and the higher number of MyoD+/MyoG−, compared to NF-YA-expressing SCs after 3 days from their activation (Fig. 4h).

Impaired differentiation ability of NF-YA$^{cKO}$ SCs is consistent with the delayed regenerative response observed in NF-YA$^{cKO}$ mice following muscle injury (Fig. 1). Indeed, after 60 days post-injury, cKO muscles still show reduced CSA and a significant increase in centrally nucleated myofibers.

To better understand which is the role of NF-Y in SCs biology, we isolated muscle stem cells and we analyzed cell cycle progression and differentiation in vitro. Despite surviving NF-YA$^{cKO}$ SCs seem to actively replicate, we observed defects in cell expansion and differentiation, associated to impaired transition from a MyoD+/MyoG+ state to a more differentiated MyoD−/MyoG+ state. Even when allowed to reach the confluence, the cells cannot activate the differentiation program efficiently (Fig. 5).

Recent Mantovani's work comes in support to the hypothesis that in muscle cells the differentiation process does not depend only on NF-YA downregulation[13], as previously demonstrated, but also on the specific NF-YA variant[16]. CRISPR-Cas9 forced deletion of exon3, which leads to NF-YAl KO and NF-YAs expression, does not result in proliferation defects but in evident differentiation impairment in C2C12 cells[16]. Besides, our previous results demonstrated that stable overexpression of NF-YAl enhances the differentiation of immortalized myoblasts that

activate a specific differentiation signature, which includes *Cdkn1C* and *Mef2D* direct transcriptional upregulation[15]. All these data are consistent with our findings that NF-YAl-depleted SCs show defects in differentiation.

RNA-seq profiles of SCs isolated from NF-YA$^{cKO}$ muscles allowed the identification of the transcriptional effects induced by NF-YA loss (Fig. 6). Cell cycle and proliferation-related terms increased following NF-YA KO, while muscle cell differentiation and skeletal muscle development categories severely decreased. The tight correlation between proliferation and differentiation processes does not allow to clearly distinguish whether the increase in cell cycle categories can be ascribed to a delay in the differentiation transcriptional program or vice versa. Surprisingly, the reduction of NF-Y binding does not affect cell cycle genes transcription, which even increases in NF-YA$^{cKO}$ cells, and therefore it seems dispensable for proliferation of muscle progenitor cells. In opposition, lowered NF-Y recruitment to 5′-UTR *Cdnk1c* and *Myogenin* E3 enhancer, regulatory regions of key genes controlling early steps of muscle differentiation, could be one of the reasons of impaired differentiation of SCs into myotubes (Fig. 5). The identification of NF-Y binding to Myogenin enhancer region can finally explain why forced NF-YAl expression strongly push immortalized myoblasts towards differentiation, even in growth culture conditions[15]. That NF-Y could regulate muscle differentiation by controlling a distal regulatory element is in agreement with R. Jothi's work in mouse ESCs: gene ontology analysis identified housekeeping functions, such as cell cycle and proliferation, as categories enriched among proximal NF-Y target genes, and genes related to embryonic development, particularly skeletal system and neuron development, among those targeted by distal NF-Y sites[35].

Further RNA-seq analysis identified GO categories related to DDR (Fig. 7). We speculate that the aberrant cell cycle progression, associated to faster DNA replication, can induce DNA damage in cultured SCs. This is consistent with our previous observations in cultured cells: NF-YA RNAi increases the overall velocity of replication forks and fork asymmetry, suggesting the occurrence of fork stalling. This eventually triggers DNA damage and cell cycle arrest[40].

An intimate relationship exists between DNA replication, DNA damage, and terminal muscle differentiation: the activation of

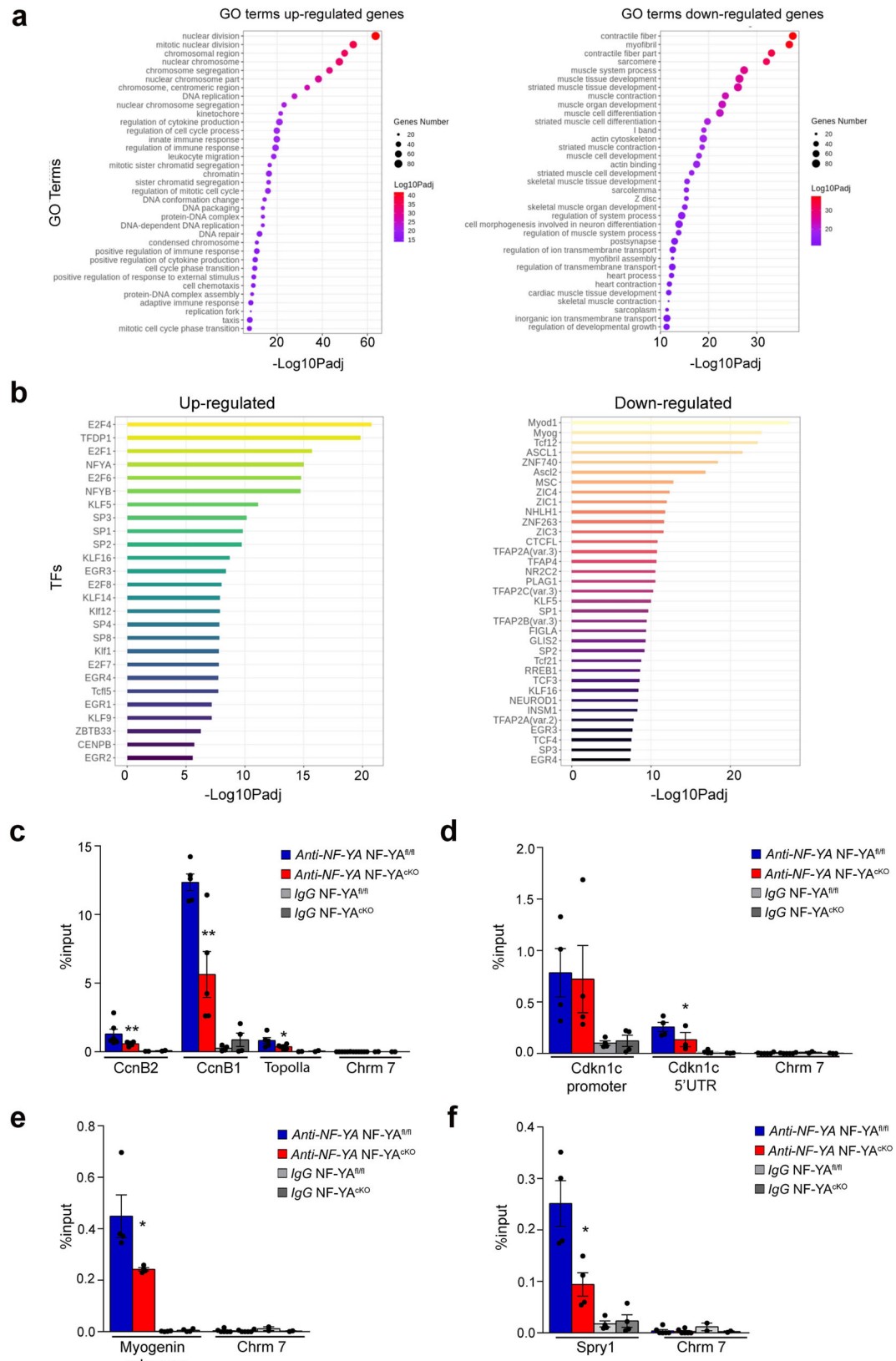

DNA damage response in replicative senescent cells or following the exposure of muscle cells to genotoxic stress interferes with the activation of the myogenic program[32,41]. Therefore, it is likely that DDR activation caused by NF-YA loss could induce the differentiation checkpoint and inhibit terminal differentiation.

Overall, we provide evidence that NF-YA expression, and hence NF-Y activity, guarantees the survival of muscle stem cells and their efficient progression from a committed to a fully differentiated state (Fig. 8). Our findings highlight how a well-known cell cycle transcriptional regulator can be repurposed in

**Fig. 6 Transcriptional effects of NF-YA KO in satellite cells. a** Enriched GO terms of up- (left) and down- (right) regulated genes in NF-YA$^{cKO}$ versus NF-YA$^{fl/fl}$ SCs. Circles are colored by adjusted $p$ value (degree of enrichment) and their size commensurate with the number of genes belonging to the relative term. **b** Enriched transcription factor motifs (TFs) in promoters of up- (left) and down- (right) regulated genes estimated with Pscan (defined as -450/+50 bp near TSS). **c** qChIP analysis of NF-YA binding to promoter regions of *CcnB2* ($p = 0.0048$), *CcnB1* ($p = 0.0055$) and *TopoIIa* ($p = 0.0265$) genes and to regulatory regions of the *Cdkn1c* gene (*Cdkn1c* 5′UTR $p = 0.0188$) (**d**) in NF-YA$^{fl/fl}$ and NF-YA$^{cKO}$ SCs. The satellite DNA-rich heterochromatin region of chromosome 7 (Chrm 7) has been used as negative control region and IgG as negative control antibody. Data are presented as mean ± s.d. (two-tailed unpaired $t$-test: *$p < 0.05$, **$p < 0.01$, $n = 5$ mice). **e** qChIP analysis of NF-YA binding to E3 enhancer of Myogenin gene ($p = 0.0480$) or to CCAAT-promoter of *Sprouty1* gene ($p = 0.0194$) (**f**). Data are presented as mean ± s.d. (two-tailed unpaired $t$-test: *$p < 0.05$, **$p < 0.01$, $n = 4$ mice).

activated stem cells to carry out key functions during myogenic differentiation.

## Methods

**Mice and animal care**. All animal experiments were carried out in compliance with national ethical regulations on the protection of animals used for scientific purposes (Italian decree no. 26 dated 04/03/2014 acknowledging European Directive 2010/63/EU). Animal procedures and protocols were approved and authorized by National Institute of Health (Ministero della Salute) (n. 404/2015-PR and no. 699/2019-PR). All the experiments involving animals were approved and supervised by the Institutional Animal Care Committee (Organismo Preposto al Benessere Animale –OPBA- University of Modena and Reggio Emilia). The Pax7-CreER$^{T2}$ mice were kindly provided to G.M. by Prof. Shahragim Tajbakhsh, Institut Pasteur, Paris[19]. NF-YA$^{fl/fl}$ mice were a gift from Nobuyuki Nukina's laboratory at Juntendo University Graduate School of Medicine (Tokyo, Japan). The NF-YA$^{fl/fl}$ mice were bred with the Pax7-CreER$^{T2}$ mice to generate NF-YA$^{fl/fl}$; Pax7-Cre ER$^{T2}$ mice, which were used as muscle SCs-specific NF-YA knockout mice (NF-YA$^{cKO}$). All animals were housed at controlled temperature and humidity on a 12 h light-dark cycle and provided water and food ad libitum. Mice were genotyped by PCR of ear DNA using genotyping Kit Phire Animal Tissue (#F140WH, Thermo Fisher Scientific) according to the manufacturer's instructions. Primers sequences used for genotyping are described in Supplementary Table 1.

**In vivo treatment**. Tamoxifen (#T5648, Merck KGaA) was dissolved in corn oil (#C8267, Merck KGaA) at the concentration of 20 mg/ml and male mice of 6–8-weeks old were injected intraperitoneally (0,1 mg/g of body weight) daily for 5 consecutive days to induce Cre-mediated deletion. Muscles of NF-YA$^{cKO}$ and relative NF-YA$^{fl/fl}$ control were harvested after 7 days from the last tamoxifen injection. To induce muscle injury, 20 μl of 10 μM Cardiotoxin (#L8102, Latoxan) dissolved in PBS was injected intramuscularly into TA muscles the day after the last Tamoxifen injection. TA muscles were collected from euthanized mice at 5, 15, 25 or 60 days post injury for histological and molecular analyses. The contralateral TA muscles were used as control for molecular analysis. To analyze cell proliferation, NF-YA$^{cKO}$ and NF-YA$^{fl/fl}$ mice were injected intraperitoneally with EdU (5 mg/kg, #C10086 Thermo Fisher Scientific) 24 h before the sacrifice.

**SCs isolation and culture**. SCs were isolated from NF-YA$^{fl/fl}$ and NF-YA$^{cKO}$ 6–8-weeks-old mice. Hindlimb muscles were dissected, mechanically cut and enzymatically digested for 1 h at 37 °C in DMEM medium containing 0.2% Collagenase I (#C0130, Merck KGaA) plus 1% Penicillin/Streptomycin. Digested muscles were mechanically disaggregated by 18-gauge needle and resuspended in manipulation medium (DMEM, 2% FBS, 1% Penicillin/Streptomycin). Cell suspension was filtered through 70 μm and 40 μm strainers, washed with manipulation medium and centrifuged for 5 min at 500 × g. SCs were then isolated using Satellite Cell Isolation Kit (#130-104-268, Miltenyi Biotec), by depletion of non-target cells through magnetical labeling with a cocktail of monoclonal antibodies (CD31, CD45, CD11b and Sca1) conjugated with MACS microbeads. NF-YA$^{fl/fl}$ and NF-YA$^{cKO}$ cells were then counted and the same number was plated on Matrigel-coated (#354248, Corning) petri with growth medium (40% DMEM, 40% HAM's F12, 20%FBS, 1% Penicillin/Streptomycin, 2.5 ng/ml human FGF-2 (#130-104-924 Miltenyi Biotec) at 37 °C, 5% CO$_2$. Cells were collected for analysis 4 days after plating. To analyze cell differentiation, growth medium was replaced with differentiation medium (DMEM, 5% horse serum, 1% Penicillin/Streptomycin) and the cells were incubated for 1, 3 or 5 days, as described in the text. To achieve the confluence, NF-YA$^{cKO}$ cells were allowed to grow for additional 2 days or were plated at higher density with respect to NF-YA$^{fl/fl}$.

**Lentivirus transduction**. Transient overexpression of NF-YAl was obtained by SCs transduction with lentiviral particles (50 MOI) for 48 h. The cDNA of NF-YAl was cloned via SmaI/SalI sites into SmaI/SalI pCCL minCMV-eGFP/hPGK-del-taLNGFR vector backbone (kind gift of Vincenzo Zappavigna, University of Modena and Reggio Emilia and Vania Broccoli, San Raffaele Scientific Institute, Milan). The resulting construct was then sequenced and used for lentiviral production. Lentiviral supernatants expressing empty vector and NF-YAl were

prepared by transfecting HEK293T packaging cells as previously described[15]. Briefly, pCCL vector plasmids (20 μg) and second generation packaging plasmids (5 μg of pMD2-VSVG and 15 μg of pCMV R8.91) were cotransfected into HEK293T cells. Lentivirus-containing supernatant was collected 24 h after transfection, centrifuged at 1700 × g to remove cell debris for 5 min, 0.45 μm-filtered and frozen at −80 °C until use.

**SCs sorting**. SCs (myoblasts) were isolated from NF-YA$^{fl/fl}$ and NF-YA$^{cKO}$ mice as described in Rossi et al.[42]. Basically, muscles were dissected, mechanically cut, and enzymatically digested at 37 °C with a solution of Collagenase I (100 mg/ml, Merck KGaA), Dispase (500 mg/ml, Gibco), and DNase I (100 mg/ml, Roche) in PBS (Merck KGaA). Undigested tissue was precipitated for 5 min, the supernatant centrifuged for 5 min at 1200 × g and cell pellet resuspended with 10% donkey serum (Merck KGaA) in PBS for 15 min at room temperature (RT). Cells were incubated 30 min on ice with primary antibody detecting SM/C-2.6 antigen, biotinylated, 1:200[43]. After extensive washes, cells were incubated with Streptavidin-APC (BD Pharmingen, 1:500) and CD45-FITC (Rat Anti-Mouse, 30-F11, BD Pharmingen, 1:100), 20 min on ice. The cells were washed again and sorted with a FACS Aria cell sorter (Beckman).

**Isolation and culture of single myofibers**. For single myofiber isolation, the EDL muscle was dissected tendon to tendon from 6- to 8-weeks-old mice and enzymatically digested in DMEM medium supplemented with 0,2% Collagenase I (#C0130, Merck KGaA) at 37 °C for 1 h. Post digestion, single myofibers were released by trituration with heat-polished glass Pasteur pipettes in manipulation medium (DMEM, 2% FBS, 1% Penicillin/Streptomycin). Floating fibers were fixed in 4% PFA after isolation (day 0, d0) or cultured in 2.0 ml tubes with no conical bottom (Eppendorf) in growth medium (40% DMEM, 40% HAM's F12, 20% FBS, 1% Penicillin/Streptomycin, 2.5 ng/ml human FGF-2 (#130-104-924 Miltenyi Biotec) for 1 (d1), 2 (d2) and 3 days (d3).

**Immunofluorescence and histology**. TA muscles were isolated from tendon to tendon, included in OCT and rapidly passed in liquid nitrogen-cooled isopentane for 1 min and left at −80 °C until processed. For histology and immuno-fluorescence analysis, 10-μm thick cross sections were cut on a cryostat (Leica). For the assessment of muscle morphology sections were stained with H&E (#460511-446644, Carlo Erba), according to standard protocols. Trichrome staining was performed by fixing sections with Bouin's solution (#429751, Carlo Erba) for 1 h and following procedures as described in ref. [44]. For immunofluorescence analysis, sections were fixed for 20 min with 10% Formalin Solution (#HT5014, Merck KGaA) (except for eMyHC staining that does not require fixation), rinsed three times with PBS for 5 min, permeabilized in methanol at −20 °C for 6 min and further rinsed two more times with PBS. For Pax7 detection, sections were subjected to the antigen unmasking treatment by boiling for 15 min in 0.01 M Na-Citrate pH 6 solution. After letting them cool, sections were blocked with PBS/4% BSA for 2–3 h, and then incubated with primary antibodies in PBS/4% BSA overnight at 4 °C. After two washes with PBS, sections were incubated with secondary antibodies (1:500, AlexaFluor Thermo Fisher Scientific) and DAPI in PBS-4% BSA (only for Pax7 this passage was preceded by the incubation with Biotin-conjugated anti-mouse IgG (1:1000, Jackson Immuno Research) in PBS/4% BSA for 45 min). For EdU and Tunel detection, Click-iT EdU cell Proliferation Kit Alexafluor 488 (Thermo Fisher Scientific) and In Situ Cell Death Detection Kit, Fluorescein (Roche) were used according to the protocol from manufacturer, respectively. After two washes with PBS, sections were mounted with mowiol. Immunofluorescences on single fibers were performed in suspension. Single myofibers were fixed for 10 min with 10% Formalin Solution (#HT5014, Merck KGaA) and washed twice with PBS. Fixed myofibers were permeabilized with PBS/0,2% Triton/50 mM NH4Cl for 10 min at RT. Myofibers were washed, blocked with PBS/2% Horse serum (Thermo Fisher Scientific) for 1 h. Primary antibodies were incubated in blocking mix over-night at 4 °C. After three washes with PBS, myofibers were incubated for 1 h with secondary antibodies (1:500, AlexaFluor Thermo Fisher Scientific) and DAPI, washed three times with PBS and mounted on slides with mowiol. Immunofluorescences on cultured SCs were performed by fixing cells for 10 min with 10% Formalin Solution (HT5014, Merck KGaA) and washed twice with PBS. The cells were then permeabilized with PBS/0,05% Triton for 10 min at RT and blocked with PBS/BSA 1% (Merck KGaA) for 1 h. Primary

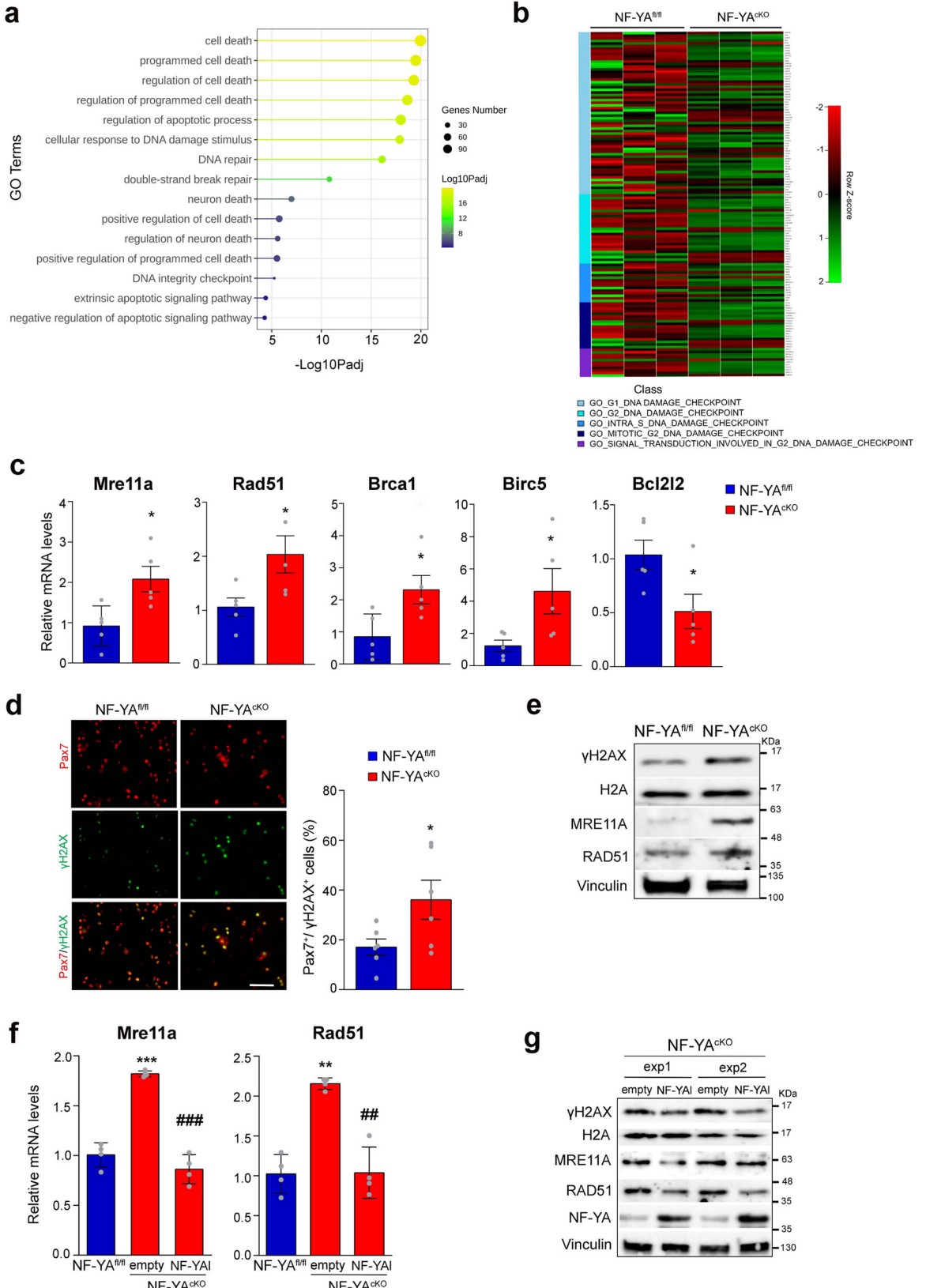

antibodies were incubated in blocking mix over-night at 4 °C in a humid chamber. After three washes with PBS, the cells were incubated for 1 h with secondary antibodies (1:500, AlexaFluor Thermo Fisher Scientific) and DAPI, washed three times with PBS and mounted on slides with mowiol. The following primary antibodies and dilutions were used: rabbit anti-NF-YA (1:100, Thermo Fisher Scientific), rabbit anti-NF-YA (H-209), (1:100, sc-10779 Santa Cruz Biotechnology), rabbit anti-laminin (1:500, ab11575 Abcam), mouse anti-Pax7 (1:100,

DSHB), mouse anti-MyHCII (MF20) (1:2, DSHB), mouse anti-eMyHC (1:50, DSHB), rabbit anti-MyoD (C-20) (1:100, sc-304 Santa Cruz Biotechnology), rabbit anti-MyoD (M-318) (1:100, sc-760 Santa Cruz Biotechnology), rabbit anti-Myogenin (M-225) (1:100, sc-576 Santa Cruz Biotechnology), mouse anti-Myogenin (1:100, M3559 Dako), anti-phospho(S139)-H2AX (1:200, #2577 Cell Signaling), rabbit anti-cleaved Caspase- 3 (D175) (1:200, #96615 Cell Signalling).

**Fig. 7 NF-YA depletion activates a DNA damage signature in satellite cells. a** Enriched upregulated GO terms (>500 genes) related to cell death/apoptosis and DNA damage/integrity processes in NF-YA^cKO *versus* NF-YA^fl/fl SCs. Circles are colored by adjusted *p* value (degree of enrichment) and their size commensurate with the number of genes belonging to the relative term. **b** Heatmaps showing expression of MSigDB genes of DNA damage checkpoints among samples. High expressed genes are highlighted in green, whereas low expressed genes in red. **c** Relative mRNA levels measured by RT-qPCR analysis of the indicated genes in SCs isolated from NF-YA^cKO versus NF-YA^fl/fl mice. Data represent mean ± s.d. (two-tailed unpaired *t*-test: *Mre11a* $p = 0.0172$, *Rad51* $p = 0.0345$, *Brca1* $p = 0.0274$, *Birc5* $p = 0.0476$, *Bcl2l2* $p = 0.0370$; $*p < 0.05$, $n = 5$ mice). **d** Immunofluorescence (left panel) images and quantification of γH2AX + cells calculated as proportion on Pax7+ cells (right panel) in proliferating SCs cultures isolated from NF-YA^fl/fl and NF-YA^cKO. Data represent mean ± s.d. (two-tailed unpaired *t*-test: $p = 0.0494$; $*p < 0.05$, $n = 6$ mice). Scale bar: 100 μm. **e** Western blot analysis of DDR markers in whole-cell extracts of NF-YA^fl/fl and NF-YA^cKO isolated SCs. Total histone H2A and Vinculin were used as loading control. $n = 3$ experiments. **f** Relative transcript levels by RT-qPCR analysis of the *Mre11* and *Rad51* genes in NF-YA^fl/fl and NF-YA^cKO SCs transduced with empty or NF-YAl lentiviral particles. Data represent mean ± s.d. (one-way ANOVA: Mre11a $F(2,9)=86.44$ $p < 0.0001$; Rad51 $F(2,9)=30.08$ $p = 0.0001$. $**p < 0.05$, $***p < 0.01$ vs NF-YA^fl/fl; $##p < 0.05$, $###p < 0.001$ vs empty). **g** Western blot analysis of DDR markers in NF-YA^cKO empty and NF-YAl-transduced satellite cells from two independent experiments (exp1 and exp2). Total histone H2A and Vinculin were used as loading control. $n = 3$ experiments.

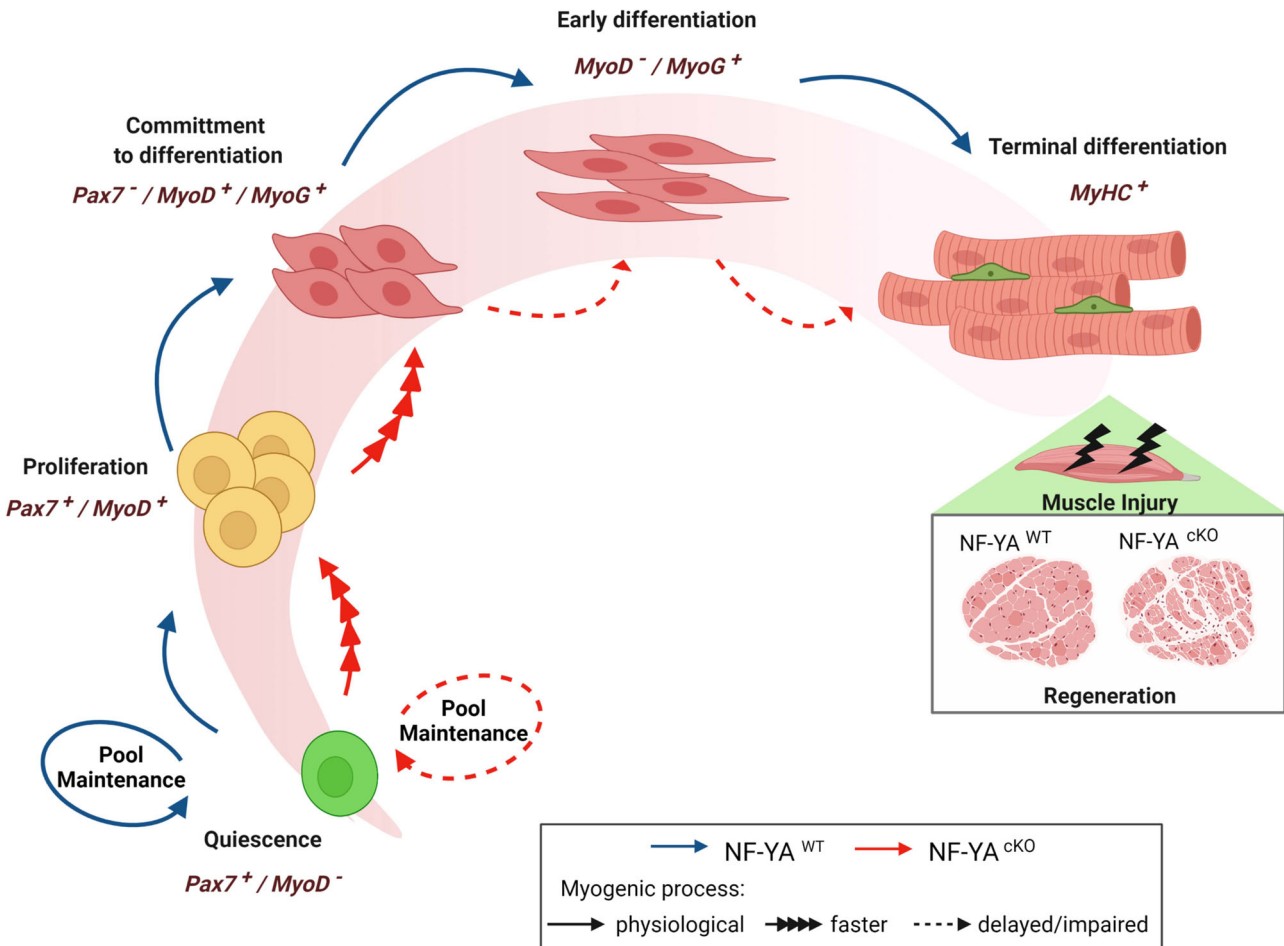

**Fig. 8 Model for NF-Y role in skeletal muscle stem cells.** Comparison between myogenic progression of SCs expressing physiological or reduced NF-YA levels (blue lines = NF-YA^WT; red lines = NF-YA^cKO). Quiescent SCs that survive to NF-YA loss (NF-YA^cKO) are quickly activated and committed to early differentiation; nevertheless, they cannot accomplish terminal differentiation. This finally affects muscle regeneration following injury in NF-YA^cKO mice. The predominant phases and related markers involved in early and late stages of SCs differentiation are indicated. Created with BioRender.com.

For EdU and Tunel detection, Click-iT EdU cell Proliferation Kit Alexafluor 488 (#C10086 Thermo Fisher Scientific) and In Situ Cell Death Detection Kit, Fluorescein (Roche) were used according to the manufacturer's protocol, respectively. Images were acquired using a Zeiss AxioSkop 40 fluorescence microscope and EVOS M5000 (Thermo Fisher Scientific) equipped with ×10 and ×20 magnification objective. CSA and infiltrated area were analyzed by ImageJ software.

**Cell cycle and proliferation analysis**. Isolated SCs or EDL myofibers were treated with EdU (10 μM, Thermo Fisher Scientific) for 2 or 20 h, respectively. For EdU detection, cells or myofibers were fixed with 3.7% PFA, washed with PBS and stained with the chemical reaction mix provided by Click-iT EdU Cell Proliferation

Assay kit (#C10086 Thermo Fisher Scientific). The cells were then immunostained with primary and secondary antibodies, as described above. Flow cytometric cell cycle analysis was performed after collecting cells by centrifugation, washed with PBS, and resuspended in 400 μL of PI solution (0.1%, Triton, 3.4 M Na citrate, 50 μg/mL Propidium Iodide). After 30 min of incubation, cells were analyzed for DNA content using cytofluorimeter (Beckman coulter)[45].

**RT-qPCR**. RNA was extracted from cells by using Ripospin II mini Kit (GeneAll), according to the manufacturer's protocol. For cDNA synthesis, 500 ng of RNA was retrotranscribed with PrimeScript RT Reagent Kit (TAKARA). Quantitative RealTime PCR was performed with SsoAdvanced Universal SYBR Green Supermix (Biorad) using Biorad CFX connect thermocycler. Oligonucleotides used were

designed to amplify 100–200 bp fragments: sequences are listed in Supplementary Table 2. The relative quantification of gene expression was calculated with the formula 2–(ΔΔCt) and normalized on the housekeeping gene Rps15.

**RNA sequencing.** RNA was quantified using Nanodrop (Thermo Fisher Scientific, Waltham, Massachusetts, USA) and quality assessment was performed by Bioanalyzer RNA 6000 nano kit (Agilent Technologies, Santa Clara, California, USA). RNAseq libraries were obtained starting from 500 ng of total RNA following Illumina TruSeq Stranded Total RNA preparation protocol. Sequencing was performed using Illumina NextSeq high-output cartridge ($2 \times 75$).

**RNA-seq data analysis.** Raw data were aligned to mouse transcriptome (mm10) through bowtie2[46]. Gene expression was quantified with RSEM 1.3.1[47]. Genes with more than 10 counts in at least two samples were kept. DESeq2 1.38 R package was employed to normalize read counts and to compute differentially expressed genes; genes with FDR < 0.01 and which Log2FC was either >1 or <−1 were considered significant. Over-represented TFs motifs in the promoter (defined as −450/+50 bp near TSS) of up- and downregulated genes were estimated with Pscan[48], employing non-redundant JASPAR 2018 set of matrices. Enriched GO Terms and Pathways, in the two gene lists, were computed by KOBAS 3.0 set for Fisher exact Test and Benjamini and Hochberg p-value correction[49]. Resulted GO Terms and Pathways with adjusted p-value lower than $10^{-5}$ were considered significant. Besides, to stand up mostly meaningful categories we discarded enriched GO Terms with background genes number higher than 500.

**Immunoblots.** Whole-cell protein extracts were prepared by resuspending cells into 1X SDS sample buffer (25 mM TrisHCl pH 6.8, 1.5 mM EDTA, 20% glycerol, 2% SDS, 5% β-mercaptoethanol, 0.0025% Bromophenol blue). Fractioned protein extracts of TA muscles were obtained homogenizing tissues in 200–500 µl of STM buffer (250 mM sucrose, 50 mM Tris–HCl pH 7.4, 5 mM $MgCl_2$, protease, and phosphatase inhibitor cocktails) for 1 min on ice using the TissueRaptor (Qiagen). The homogenate was maintained on ice for 30 min, vortexed and centrifuged at 800 g for 15 min. The supernatant (cytosolic fraction) was collected while the pellet was washed with STM buffer, centrifuged, and resuspended in 200–500 µl NET buffer (20 mM HEPES pH 7.9, 1.5 mM $MgCl_2$, 0.5 M NaCl, 0.2 mM EDTA, 20% glycerol, 1%Triton-X-100, protease and phosphatase inhibitors). The nuclear fraction was vortexed, incubated on ice for 30 min, and lysed with 10 passages through an 18-gauge needle. The nuclear lysate was centrifuged at $9000 \times g$ for 30 min, at 4 °C, the resultant supernatant represented the soluble nuclear fraction[50]. For immunoblotting equivalent amounts of cellular extracts were resolved by SDS-PAGE, electrotransferred to PVDF membrane (GE Healthcare) and immunoblotted with the following primary antibodies diluted in TBS 1X- BSA 1 µg/µl: mouse anti-NF-YA (G2) (1:1000, sc-17753, Santa Cruz Biotechnology), mouse anti-Pax7 (1:1000, DSHB), mouse anti-MyHCII (MF20) (1:50, DSHB), rabbit anti-Myogenin (M-225) (1:1000, sc-576 Santa Cruz Biotechnology), mouse anti-eMyHC (1:50, DSHB), rabbit anti-MyoD (C-20) (1:1000, sc-304 Santa Cruz Biotechnology), rabbit anti-MyoD (1:1000, 18943-1-AP Proteintech), rabbit anti-phospho(S10)-histone H3 (1:1000, #06-750 Millipore), goat anti-histone H3 (1:1000, sc-8654 Santa Cruz Biotechnology), rabbit anti-MRE11A (1:1000, E-AB-11403 Elabscience), rabbit anti-RAD51 (H-92) (1:1000, sc-8349 Santa Cruz Biotechnology), rabbit anti-phospho(S139)-H2AX (1:1000, #2577 Cell Signaling), rabbit anti-histone H2A (1:1000, Ab18255 Abcam), goat anti-Actin (I-19) (1:5000, sc-1616, Santa Cruz Biotechnology), mouse anti-Vinculin (V4504, Merck KGaA), rabbit anti-GAPDH (1:1000, sc-47727 Santa Cruz Biotechnology), mouse anti-Tubulin (1:1000, 66031 Proteintech Europe), mouse anti-p57Kip2 (66794, Proteintech Europe), rabbit anti-NF-YB (GeneSpin). Membranes were blotted and scanned with Amersham Imager AI680 RGB (GE Healthcare), using chemiluminescent detection reagents Westar ηC and Supernova HRP substrates (Cyanagen).

**Chromatin immunoprecipitation.** Cultured SCs were incubated for 10 min with 1% formaldehyde; after quenching the reaction with 0.1 M glycine and nuclear fraction was sonicated to obtain 500–800-bp DNA fragments. The chromatin solution was precleared by adding protein G-agarose (#5720-0002 KPL) for 2 h at 4 °C, aliquoted and incubated with 4 µg of anti-NF-YA (#C15310261, Diagenode) or rabbit IgG (sc-2027, Santa Cruz Biotechnology) overnight at 4 °C on a rotating wheel. Before use, protein G-agarose was blocked twice at 4 °C with 1 µg/µl salmon sperm DNA sheared at 500 bp length and 1 µg/µl BSA for 2 h. After washes of the protein G-agarose, DNAs were isolated by phenol-chloroform extraction and resuspended in TE buffer[15]. Quantitative Real-Time PCR was performed using with SsoAdvanced Universal SYBR Green Supermix (Biorad) in the Biorad CFX connect machine. Sequences of primers used are listed in Supplementary Table 3.

**Statistical analysis.** All data shown in graphs represent the mean of at least three independent experiments or mice ± s.d. values. All statistical analyses were performed using GraphPad PRISM 6.0 software using one-way ANOVA or two-tailed unpaired Student's t-test, as appropriate. p values of $p < 0.05$ were considered to be statistically significant (*), $p < 0.01$ (**), $p < 0.001$ (***), and $p < 0.0001$ (****), n.s.: not significant ($p \geq 0.05$).

**Reporting summary.** Further information on research design is available in the Nature Research Reporting Summary linked to this article.

## Data availability
RNA-seq data generated in this study have been deposited in the Gene Expression Omnibus (GEO) database under the accession code GSE154017. All raw data generated in this study are provided in the Source Data file. Source data are provided with this paper.

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

## Acknowledgements

This work was supported by The French Muscular Dystrophy Association (AFM-Téléthon) Trampoline Grant (n.16408) and Research Grant (no. 18364) to C.I. and R.M., by FAR_DIP2017- Department of Life Science, University of Modena and Reggio Emilia to C.I., and by the Italian Ministry of Health Grant GR-2013-02355625 to D.D. The laboratory of C.I. is supported by AIRC IG 2018 no. 21323. We thank Antonella Franchini (the University of Modena and Reggio Emilia) for sharing reagents and Antonio Musarò (Sapienza University of Rome) for helpful discussion.

## Author contributions

G.R. designed and performed the experiments. V.B., S.B., L.L., M.S. and G.M. performed the experiments. A.C. supervised gene expression profiling experiments. A.V. provided reagents. D.D., M.R. and E.S. performed bioinformatic analyses. S.M., L.L., G.M. and R.M. provided critical revision of the article. C.I. conceptualized the study, supervised the research, designed experiments, and wrote the paper.

## Competing interests

The authors declare no competing interests.
