## [Peer Review File · Nature Communications]

Reviewers' Comments:

Reviewer #1:

Remarks to the Author:

This series of experiments investigated the role of the transcription factor, NF-Y, in driving muscle stem cell fate in skeletal muscle at rest and during regeneration. Many of the conclusions are based on a muscle stem cell specific NF-YA inducible knockout. A combination of in vitro and in vivo experiments were conducted with very interesting results. The authors demonstrate that satellite cells express the NF-YA long isoform, which appears to preferentially impact cell differentiation, which is in contrast to the role of NF-YA in embryonic and hematopoietic stem cells, where it appears to regulate cell proliferation. This is quite an interesting finding. Overall, this is very nice paper, the conclusions of which are generally well supported by the data. There are several points that require some consideration outlined below.

Specific comments:

1. The data in figure 1 demonstrates that regeneration of skeletal muscle following injury is delayed in the absence NF-YA. H&E stains and measurement of CSA appears convincing. The message is that up until 25d regeneration in the NF-YA conditional KO are delayed. I think an important question is whether the muscle ever regenerates to normal levels. If the time course were taken out further would CSA in the KO animals reach similar levels to that of the control animals? Additionally, is muscle function also impaired? These are important questions as it speaks to the biological redundancy of NF-YA in driving repair.
2. The statement on line 130-131 is misleading. Even in the injured KO animals Pax7 went up, albeit not to a statistically significant level. It is also interesting that the western blot data does not support the PCR data with respect to Pax7 at all.
3. Given the results of Cui and colleagues (2020). Is it possible that NFE2L1 can pick up the role of NF-YA and successfully complete the regenerative process, albeit in a delayed fashion? Did the authors consider examining NFE2L1 in the context of the experiments presented?
4. Figure 4E, F – these results should be expressed as a proportion of activated cells. If fewer cells are activated to begin with then it stands to reason that there will be fewer differentiating cells. In my opinion, this data as presented is not that informative.
5. It is interesting that NF-Y appears important for proliferation of embryonic and hematopoietic stem cells but not satellite cells, where NF-Y appears more important in driving differentiating. Is it possible that this is due to the multipotent nature of satellite cells as compared to the pluripotent and multipotent nature of embryonic and hematopoietic stem cells? This may be worth raising.
6. Line 295-297. It isn't clear to me what is being inferred here. "failure of SCs to expand, prior to the onset of activation....". SCs do not expand prior to activation. In fact, activation is a necessary step in expansion of the population.

Reviewer #2:

Remarks to the Author:

Comments to authors:

This manuscript uses a conditional mouse model to explore the role of NF-Y in regulation of satellite cell pool. This is a follow up story to the previously published work by the same group describing different functions of NF-Y isoforms (long and short isoforms) in regulation of cell growth and differentiation.

In general the experiments are poorly designed, as will be described below and mechanistic insights into which isoform of NF-Y regulates satellite cell numbers and how it does so is lacking. The authors claim that NF-YA is needed to maintain quiescence and point to a bunch of cell cycle inhibitor genes and other key genes in figure 6 that show they are reduced in KO, but that also MyoD is reduced and that Pax7 is increased, which to me it shouldn't be if they are breaking quiescence. In fact, if quiescence is lost, then MyoD should be up and the cell cycle genes should

be up and therefore the # of SC should actually be higher in the KO, they can't go back to quiescence or differentiate sure, but they should still be proliferating throughout. So I want to ask the authors how they justify their findings, basically why are the cells dying instead of proliferating in vivo? Because in culture they grow just fine. Also what I don't understand is that under normal circumstances NF-YA goes down in differentiation, so why does the loss of NF-YA stop differentiation? Shouldn't it improve differentiation?

Specific comments:

1. Is the long isoform of NF-Y really important for sat cells and how?, or if the short isoform would work if the long one is absent. I would suggest the authors make a stable cell line of the KO cells with either the short or the long isoform reintroduced and then perform RNA-Seq analysis to differentiate the function of long versus short isoform of NF-Y.
2. Do a count of the percentage of centrally located nuclei in injured WT and KO muscle, see if regeneration is delayed.
3. Do a count of the total number of SC on isolated EDL fibers in KO and WT
4. Show how the SC in the KO are lost, so far there's nothing that explains the loss of SC, except for a tunel assay. They have one tunel assay that shows most of the KO SC dying. Show it through other methods like cleaved caspase 3. I would also stain the TA cross sections for MyoD to see if there's a # of MyoD+/Pax7- cells to remove the possibility that the SC have broken quiescence and are proliferating without being able to differentiate. Lastly, do a rescue experiment with either the long or the short NF-Y isoform in satellite cells of KO mice.
5. Follow up on 4, but inject live mice with EDU and then stain TA cross sections. If SC are breaking quiescence then they should be incorporating more EDU in the KO
6. Follow up on 4. If the SC are dying then in the RNA-seq data show that apoptotic genes are upregulated. Or find some genes that would explain the loss of SC
7. Redo the differentiation assay with more confluent cells and for a longer period of time to see if there's just a delayed activation. (because based on the figure pictures the KO were not nearly confluent enough to differentiate)
8. In Figure 1, the authors have done western blot on whole muscle and draw conclusion for the expression in sat cells based on whole muscle analysis. This conclusion is invalid as muscle is a heterogenous tissue and contain numerous non-myogenic cells.
9. Figure 2B: Claims that SC are dying or breaking quiescence yet show even higher pax7 expression in KO under normal conditions, wouldn't that imply that they are actually more quiescent?. Also the eMyHC (same with myogenin, myf5 does look real to me) is not convincing to me, they shouldn't be comparing it to steady state, it should be control vs KO in both states. Also, they are using S.E.M and not Standard deviation. I understand that if they used the WT as 1 in both conditions it would be misleading, but seeing ¼ reduction is not very convincing and it only looks cool when you go 200 vs 150, instead of 1 vs 0.75
10. Figure 2C: through staining that there is a marked expression of eMyHC at 5 days (and say that WT has a more defined staining but I see no difference between, but in WB there is clearly less in the KO.)
11. Figure 2D: really bad western blot.
12. Figure 3A: Those fiber figure is low quality. And I fail to see the point of it, in all conditions NF-YA is present at all times, it hardly screams that it has a key function important for quiescence and differentiation if it is always present. They claim in the later panels that NF-YA is greatly reduced in differentiation, but the staining really doesn't show that.
13. Figure 4B: Change the y-axis, # of pax7+ cells/fiber is wrong for cross sections. Just do # of Pax7+/mm²
14. Figure 4E: the panel here is misleading. They don't say at what time post isolation this was done and it would be better represented as a percentage, not a total number as they establish in the previous panels that the KO has less cells to begin with. Also, I would like to see a total number of SC per fiber as confirmation for previous panels.
15. Figure 4F: Same as figure 4E needs to be percentage not total number as KO has less cells. Also the staining pictures are very low quality and need to be their own panel in my mind.
16. Figure 5 A,B and C don't line up. A and B are claiming that % of Pax7 cells are lower in KO but C is saying the complete opposite that it's actually higher in the KO.
17. Figure 5 D is strange data, they only added EDU for 2 hours, there's no way they got 40% positive cells in WT in such short time. Also the time EDU was added is not stated in the figure

legend but it is in the materials and methods.

18. Figure 5 F is pointless in my mind. It's just the % of myogenin cells in normal growth culture.

19. Figure 5 G: It would be nice if they left it in DM for longer to see if eventually the KO will differentiate properly.

20. Figure 7B/D: that break in the y axis is completely unnecessary and is only there to make it look like there's a bigger difference than there is.

We thank the Reviewers for very constructive suggestion. We addressed their concerns accordingly. As suggested by the two expert Reviewers, we performed new experiments that allowed us to improve the quality of the manuscript.

Below we present our point-to-point response. The amendments are highlighted in red color in the revised manuscript.

Reviewer #1 (Remarks to the Author):

This series of experiments investigated the role of the transcription factor, NF-Y, in driving muscle stem cell fate in skeletal muscle at rest and during regeneration. Many of the conclusions are based on a muscle stem cell specific NF-YA inducible knockout. A combination of in vitro and in vivo experiments were conducted with very interesting results. The authors demonstrate that satellite cells express the NF-YA long isoform, which appears to preferentially impact cell differentiation, which is in contrast to the role of NF-YA in embryonic and hematopoietic stem cells, where it appears to regulate cell proliferation. This is quite an interesting finding. Overall, this is very nice paper, the conclusions of which are generally well supported by the data.

We thank the reviewer for encouraging comments and forthcoming critical comments.

Specific comments:

1. The data in figure 1 demonstrates that regeneration of skeletal muscle following injury is delayed in the absence NF-YA. H&E stains and measurement of CSA appears convincing. The message is that up until 25d regeneration in the NF-YA conditional KO are delayed. I think an important question is whether the muscle ever regenerates to normal levels. If the time course were taken out further would CSA in the KO animals reach similar levels to that of the control animals? Additionally, is muscle function also impaired? These are important questions as it speaks to the biological redundancy of NF-YA in driving repair.

According to the Reviewer's suggestion, we performed additional *in vivo* experiments. We analyzed cardiotoxin-treated muscles of control and NF-YA^{CKO} mice following 60 days from injury. NF-YA^{CKO} mice showed reduced TA muscle weight after cardiotoxin-damage compared to NF-YA^{fl/fl}, suggesting defects in muscle regeneration even at longer time (new Suppl. Figure 1F). No differences were observed in the weight of vastus and gastrocnemius undamaged muscles from the same mice that received cardiotoxin injection in TA, supporting the absence of alterations in NF-YA^{CKO} muscle mass in physiological condition (Suppl. 1E). H&E staining highlighted smaller fiber cross-sectional area and higher presence of centrally nucleated fibers in the KO animals even upon 60 days (new Fig. 1F-I). Despite this, muscle function was not impaired, at least when determined by the forelimb grip strength test, a widely used conventional and not invasive method to assess skeletal muscle function in rodents (new Suppl. Figure. 1G).

These new results have been added to the manuscript.

2. The statement on line 130-131 is misleading. Even in the injured KO animals Pax7 went up, albeit not to a statistically significant level. It is also interesting that the western blot data does not support the PCR data with respect to Pax7 at all.

We agree that the difference between Pax7 transcript and protein levels observed in NF-YA^{CKO} mice is an interesting but open question.

RNA-seq analysis of SCs isolated from NF-YA^{CKO} mice shows the same result, highlighting that NF-YA decrease is associated to increased Pax7 gene transcription but reduced protein levels. This suggests that Pax7 could be post-transcriptionally controlled in NF-YA^{CKO} SCs by indirect mechanisms.

Transcriptional activation of the Pax7 gene is not the unique mechanism required to increase Pax7 protein¹. It has been demonstrated that Pax7 protein levels are directly regulated by a fine balance

between Nedd4-mediated ubiquitination and casein kinase 2-dependent phosphorylation at S201^{2,3}. Therefore, these post-translational mechanisms could be perturbed in NF-YA^{cKO} muscle leading to Pax7 protein decrease, which in turn could trigger a feedback mechanism on Pax7 transcription in an attempt to restore normal protein levels.

We hope that the Referee will agree with us that the mechanism underlying transcription and post-translational control of Pax7 will require several efforts that we envisage to take on in our future research.

The aim of RT-qPCR analysis shown in Figure 2B is to determine whether also NF-YA^{cKO} SCs are able to activate the transcription of key myogenic genes following muscle damage, as NF-YA^{fl/fl} cells do. As suggested by Referee 2, we decided to show gene expression of CTX-samples relative to untreated-ones, which have been arbitrarily set at 1 in both models¹¹ (new Figure 2B).

3. Given the results of Cui and colleagues (2020). Is it possible that NFE2L1 can pick up the role of NF-YA and successfully complete the regenerative process, albeit in a delayed fashion? Did the authors consider examining NFE2L1 in the context of the experiments presented?

We addressed this very interesting question with our available tools. We identified a significant decrease in transcription of the Nfe2l1 gene (log2FC=-0.55, pvalue=0.01), which was confirmed by RT-qPCR analysis (Figure 1A-Referee 1). Therefore, it is possible that both NF-YA and NFE2L1 are involved in preserving satellite cells proliferation and viability. This would require rescue experiments in NF-YA^{cKO} mice, which we could not perform because not described within the document approved for animal manipulation by the Italian Ministry of Health (n. 404/2015-PR and 699/2019-PR). The promoter region of the murine *Nfe2l1* gene contains multiple CCAAT boxes, suggesting a potential regulation of *Nfe2l1* transcription by NF-Y itself (Figure 1B-Referee 1). qChIP analysis with anti-NF-YA and the highly efficient anti-NF-YB antibodies corroborated that NF-Y is associated to the regulatory regions of Nfe2l1 gene, in isolated satellite cells (Figure 1C-Referee 1). Despite not significant, we identified a decrease in NF-Y binding in satellite cells isolated from NF-YA^{cKO} mouse muscles compared to control ones (Figure 1D-Referee 1).

Further research is needed, *in vivo* and *ex vivo*, to better understand genetic and functional interaction between NF-Y and Nfe2l1. We decided not to include these results in the manuscript and we hope that the Referee will agree with our choice.

A

	log2FoldChange	padj
Nfe2l1	-0,55834807	0,015068

**B****C****D**
Figure 1-Referee 1: A. Expression levels of the *Nfe2l1* gene in isolated SCs by means of RNA-seq analysis (upper panel) and RT-qPCR assay (lower panel). Unpaired Student t-test: * $p < 0.05$. B. Schematic representation of the *Nfe2l1* murine gene (UCSC Genome browser). The encircled bars represent the CCAAT elements in the upstream regulatory region. C. ChIP analysis of NF-YA and NF-YB binding to *Nfe2l1* regulatory region in SCs isolated from wt mice. One-way ANOVA: ** $p < 0.01$, **** $p < 0.0001$ vs IgG. D. ChIP analysis of NF-YA binding to *Nfe2l1* regulatory region in control NF-YA^{fl/fl} and NF-YA^{cKO} SCs.

4. Figure 4E, F – these results should be expressed as a proportion of activated cells. If fewer cells are activated to begin with then it stands to reason that there will be fewer differentiating cells. In my opinion, this data as presented is not that informative.

We calculated the relative percentages of myogenic populations identified by immunofluorescence and we changed the Figures, accordingly (new Figures 4E, H and I).

5. It is interesting that *NF-Y* appears important for proliferation of embryonic and hematopoietic stem cells but not satellite cells, where *NF-Y* appears more important in driving differentiating. Is it possible that this is due to the multipotent nature of satellite cells as compared to the pluripotent and multipotent nature of embryonic and hematopoietic stem cells? This may be worth raising.

Thanks to Reviewers suggestion, we deepen our analysis on the effects of NF-YA abrogation *in vivo* and *ex vivo*. In muscle section of uninjured mice, the number of Pax7+ cells significantly decreased. Detection of Tunel+/Pax7+ and cleaved-Casp3+/Pax7+ cells indicates that cell death occurs in satellite cells following NF-YA loss (new Figures 4C and D). Moreover, the relative distribution of myogenic cells identified by immunofluorescence showed a significant decrease in Pax7+/MyoD- quiescent and Pax7+/MyoD+ activated cells and a concomitant increase in early differentiating Pax7-/MyoD+ cells (new Figure 4E). In parallel, we analyzed the percentage of Pax7+/EdU+ cells in TA sections, which significantly decreased (new Figure 4F). These data represent a freeze frame of the effects of NF-YA depletion on muscle homeostasis, with the main consequence being the loss of Pax7+ cells and the depletion of quiescent and activated cells.

To further understand how NF-YA abrogation affects SCs fate, we analyzed isolated EDL myofibers cultured in suspension, which allow to observe the kinetics of activation, division and differentiation of SCs (new Figure 4H). At time 0 h, the number of Pax7+ cells/myofiber was significantly hit, consistently to what observed in TA sections (new Figure 4G). Despite not statistically significant, we observed changes in cell distribution within Pax7+ cells of NF-YA^{CKO} fibers, with the percentage of Pax7+/MyoD- quiescent cells decreasing in favor of Pax7+/MyoD+ proliferating cells. This hints that NF-YA^{CKO} SCs are prone to be activated earlier than control cells. Following 24h of culture, quiescent SCs were almost lost in both wt and NF-YA^{CKO} fibers. We detected an higher percentage of Pax7-/MyoD+ cells committed to differentiation in NF-YA^{CKO} compared to wt (62.3% versus 53.4%), at the expense of proliferating Pax7+/MyoD+ cells (34% versus 41%). At 48h, the majority of MyoD+ NF-YA^{CKO} cells already expressed the early differentiation marker Myogenin (65% versus 48%). Finally, at 72h, the distribution of NF-YA^{CKO} seems similar to the previous time point: MyoD+/MyoG+ cells did not increase (67.1%) and about 30% of MyoD+/MyoG- population was still detected (compared to 34.3% at 48h). Only 3% of NF-YA^{CKO} were able to proceed towards the next differentiation step (MyoD-/MyoG+) compared to 14.3% of control cells, these showing a significant decrease in proliferating population (MyoD+/MyoG-) when compared to 48h time point (from 51.2% to 18.8%). This time course suggested that NF-YA down-regulation push the cells to exit the quiescence and enter the cell cycle. Once activated, NF-YA loss stimulates the cells to quickly undertake the early myogenic program, which cannot proceed towards a complete differentiation.

Finally, to rule out the possibility that NF-YA loss stimulates SCs cells to break the quiescence and directly differentiate, bypassing DNA replication, we performed an EdU pulse and release experiment in EDL myofibers cultured in suspension. By monitoring EdU+ cells, we observed a robust increase in EdU+/Pax7- cells percentage in NF-YA^{CKO} fibers, consistently with a rapid commitment to differentiation (new Figure 4I). Moreover, at 72h post-isolation, we were not able to identify MyoG+ cells negative for EdU either in wt or NF-YA^{CKO} fibers, demonstrating that all differentiated cells previously underwent DNA replication.

Also in SCs isolated and cultured *in vitro*, NF-YA depletion hit the number of Pax7+ cells, but the analysis of surviving SCs shows an increase in the percentage of Pax7+/MyoD+ and Pax7+/EdU+

cells (Figures 5A, B and C). These data could be suggestive of an increase of the proliferating population, but time course analysis of SCs number in growth condition did not support this interpretation (new Figure 5F). Cell cycle analysis showed an increase in the G2/M population that did not match with an increase in phospho(Ser10)-H3 mitotic levels (new Figure 5E). This new result could be suggestive of an impairment in cell cycle progression following NF-YA abrogation, as we observed in other cell types⁴. Similarly to what we previously demonstrated, an increase in γ H2AX+ cells was detected in NF-YA-depleted (new Figures 7D and E). In addition to a decrease in myogenic categories, RNA-seq profile of NF-YA^{CKO} SCs identified an enrichment in DDR (DNA Damage Response) GO terms, which we validated through RT-qPCR of selected genes (new Figures 7A-C). These results are consistent with data describing that DDR interferes with the activation of the myogenic program⁵⁻⁷.

Overall, our new results highlight that NF-YA decrease addressed the cells towards differentiation. If NF-YA is reduced too early, the cells cannot properly exit the cell cycle and accumulate DNA damage, which interferes with a correct differentiation program.

The presence of the NF-YA1 isoform, which is usually observed in undifferentiated cells, could confer to NF-Y not only the ability to properly complete cell cycle progression of the adult muscle stem cell population, but also to correctly activate the differentiation program (new Figures 5F and G). CRISPR/Cas9 genome editing targeting exon3 could allow in the future to better identify the role of NF-YA1 *versus* NF-YAs in the fate of SCs. These new results and considerations have been added in the manuscript.

6. Line 295-297. *It isn't clear to me what is being inferred here. "failure of SCs to expand, prior to the onset of activation.....". SCs do not expand prior to activation. In fact, activation is a necessary step in expansion of the population.*

We are sorry for this inaccuracy. We modified the text, accordingly.

Reviewer #2 (Remarks to the Author):

In general the experiments are poorly designed, as will be described below and mechanistic insights into which isoform of NF-Y regulates satellite cell numbers and how it does so is lacking.

We thank the Referee for the careful and insightful review of our manuscript. We respect Referee's opinion and concerns on our work. We are aware that our previous version of the manuscript did not allow to get a clear conclusion on the activity of NF-Y in muscle stem cells. We addressed all the concerns raised by the Referee and we performed new experiments to clarify the mechanisms through which NF-Y controls SCs fate. We hope that the Referee will find our new results and responses to her/his comments satisfactory.

The authors claim that NF-YA is needed to maintain quiescence and point to a bunch of cell cycle inhibitor genes and other key genes in figure 6 that show they are reduced in KO, but that also MyoD is reduced and that Pax7 is increased, which to me it shouldn't be if they are breaking quiescence. In fact, if quiescence is lost, then MyoD should be up and the cell cycle genes should be up and therefore the # of SC should actually be higher in the KO, they can't go back to quiescence or differentiate sure, but they should still be proliferating throughout. So I want to ask the authors how they justify their findings, basically why are the cells dying instead of proliferating in vivo? Because in culture they grow just fine.

Thanks to this comment, we decided to perform additional experiments. As suggested by the Referee (points 4 and 5), we injected live mice with EdU, we stained TA cross sections and we followed SCs expansion through Pax7/MyoD and MyoG staining in isolated EDL myofibers. These new results showed that NF-YA loss reduces the survival of SCs *in vivo* and, once activated, surviving SCs quickly proliferate and are committed to differentiation, despite delayed in proceeding throughout the differentiation program (Figure 4). *In vitro*, these cells seem to quickly

replicate DNA because of the increase in EdU+/Pax7+ cells (Figure 5C). Despite this, NF-YA^{CKO} cell culture cannot properly expand, as highlighted by the analysis of phospho(Ser10)-H3 expression and cell confluence (new Figures 5F and G). These defects in cell cycle progression eventually triggers the activation of the DNA damage response, which could have as a consequence the activation of the differentiation checkpoint (new Figure 7). Below, we addressed all these concerns point by point (see points 4, 5, 6 and 7).

Also what I don't understand is that under normal circumstances NF-YA goes down in differentiation, so why does the loss of NF-YA stop differentiation? Shouldn't it improve differentiation?

Western blot analysis previously published did not detect NF-YA expression in whole extracts from muscle tissues, composed by mature muscle fibers deriving from the fusion of myoblasts. Moreover, transient overexpression of NF-YA impairs the differentiation of immortalized C2C12 myoblasts⁸. This has led for many years to the idea that NF-YA down-regulation is required for proper muscle differentiation. Unexpectedly, stable NF-YA overexpression brought to light new results: while NF-YAs forced expression stimulates cell proliferation and impairs differentiation, NF-YA1 enhances the activation of the differentiation program⁹. These results opened a new scenario in which NF-YA1 could participate to cell differentiation program.

We previously showed that NF-YA levels progressively decrease with differentiation of immortalized myoblasts, but NF-YA is still detectable during early differentiation, when Myogenin is expressed⁹. In agreement with this, immunofluorescence staining of SCs retained within myofibers shows the presence of NF-YA+/MyoG+ co-stained cells (Figure 3A). To further support these results, we performed a time-course analysis of NF-YA protein levels in SCs during differentiation (new Figure 3C).

Overall, these data unequivocally demonstrate that NF-YA is still expressed in early differentiating cells. Moreover, the identification of NF-Y binding to Myogenin enhancer suggests that NF-Y could have a direct function in the differentiation program (Figure 6E).

Our new results, obtained in isolated myofibers, highlight that NF-YA^{CKO} SCs quickly proliferate and are committed to early differentiation (Pax7-/MyoD+ cells), but are severely delayed in the progression throughout the differentiation program (new Figure 4H). At 0h from myofibers isolation, we observed a lower proportion of Pax7+/MyoD- quiescent cells and a higher percentage of Pax7+/MyoD+ proliferating cells in NF-YA^{CKO}, with respect to control fibers. Upon 24h, NF-YA^{CKO} myofibers have a higher percentage of Pax7-/MyoD+ cells committed to differentiation compared to control (62.3% versus 53.4%), associated to a lower proportion of proliferating Pax7+/MyoD+ cells (34% versus 41%). After 48h, 65% of NF-YA^{CKO} cells already express Myogenin, compared to 48% of control cells. At 72h, MyoD+/MyoG+ cells do not increase (67.1%) and 29.9% of MyoD+/MyoG- population is still detected in NF-YA^{CKO} (compared to 34.3% at 48 h). The delay in the myogenic differentiation program is definitely highlighted by the robust difference between the percentage of MyoD-/MyoG+ in NF-YA^{CKO} (3%) compared to control cells (14.3%) after 72h. This is associated to high percentage of cells still proliferating (MyoD+/MyoG-) in NF-YA^{CKO} compared to the previous time point, differently from control cells that show a decrease of MyoD+/MyoG- population from 51.2% to 18.8%.

Overall, time course assay showed that NF-YA loss stimulates the cells to quickly proliferate and activate the early myogenic program, without being able to proceed towards a complete differentiation.

SCs isolated from NF-YA^{CKO} mice and cultured *in vitro* show that NF-YA loss increases DNA replication but accumulate DNA damage, which is known to inhibit muscle cell differentiation⁷ (new Figure 5 and Figure 7). Therefore, it is not surprising that NF-YA loss has clear consequences on cell differentiation by indirect and direct transcriptional mechanisms.

These new results have been added in the manuscript.

Specific comments:

1. Is the long isoform of NF-Y really important for sat cells and how? or if the short isoform would work if the long one is absent. I would suggest the authors make a stable cell line of the KO cells with either the short or the long isoform reintroduced and then perform RNA-Seq analysis to differentiate the function of long versus short isoform of NF-Y.

This is an interesting question that we would like to address through genome editing techniques to obtain clones expressing exclusively NF-YAs and compare them to wt NF-YA1 expressing cells. Unfortunately, we were not able to isolate by fluorescent-activated cell sorting (FACS) a number of skeletal muscle stem cells suitable for CRISPR/Cas9 genome editing technique (as already done with C2C12 immortalized myoblasts)¹⁰. This represents our next challenge that will allow to gather functional information on the role of the two isoforms by sequencing the whole transcriptome of the two isogenic cell populations. Anyway, we tried to overexpress NF-YAs or NF-YA1 by lentiviral transduction of NF-YA^{ckO} SCs isolated and cultured *in vitro*. Although the percentage of Pax7+ cells that survive *in vitro* was dramatically reduced by NF-YA inactivation, NF-YAs and NF-YA1 overexpression was achieved in SCs. RT-qPCRs were performed on selected target genes to identify whether the two isoforms were equally able to rescue transcription of myogenic genes (the majority of which are not NF-Y direct target genes) (Figure 2- Referee 2). A significant increase in MyoD transcription was induced by NF-YA1. Despite not significant, also the transcription of MyoG, Mef2C and Cdkn1A (p21) genes were induced. Surprisingly, NF-YAs represses CyclinA2 and CyclinB2 expression, which could be a consequence of the release from G2/M arrest.

We are well aware of the limitation of this experiment, since isolation and culture pressure selected only a subpopulation of NF-YA^{ckO} SCs which does not completely resemble the *in vivo* scenario. Therefore, we believe that these results cannot give a clear insights into the role of the two isoforms, which we hope to obtain through CRISPR/Cas9 genome editing. We decided not to include these results in the manuscript and we hope that the Referee will agree with our choice.

Figure 2-Referee 2: A. RT-qPCR analysis of NF-YA isoforms in NF-YA^{CKO} satellite cells transduced with Empty, NF-YAI and NF-YAs lentiviral particles. B. RT-qPCR analysis of myogenic and cell cycle genes in transduced NF-YA^{CKO} satellite cells. The expression levels in Empty cells have been arbitrarily set at 1. One way ANOVA: **p<0.01, ***p< 0.001.

2. *Do a count of the percentage of centrally located nuclei in injured WT and KO muscle, see if regeneration is delayed.*

We added the count of centrally nucleated fibers, accordingly (new Figure 1H).

3. *Do a count of the total number of SC on isolated EDL fibers in KO and WT.*

The count of SCs on isolated fibers was already included in the previous version of the manuscript (Figure 4B, now Figure 4G). We added the histogram representing the number of Pax7+ cells/mm² in TA sections (new Figure 4A, right panel).

4. *Show how the SC in the KO are lost, so far there's nothing that explains the loss of SC, except for a tunel assay. They have one tunel assay that shows most of the KO SC dying. Show it through other methods like cleaved caspase 3. I would also stain the TA cross sections for MyoD to see if there's a # of MyoD+/Pax7- cells to remove the possibility that the SC have broken quiescence and are proliferating without being able to differentiate. Lastly, do a rescue experiment with either the long or the short NF-Y isoform in satellite cells of KO mice.*

We performed the immunofluorescence analysis of cleaved-caspase3 on TA sections, as suggested by the Referee (new Figure 4D).

We also stained TA cross sections with MyoD and Pax7 (new Figure 4E): we observed a significant decrease in both Pax7+/MyoD- and Pax7+/MyoD population, associated to a robust increase in Pax7-/MyoD+ cells. As described above (see point 5, Referee 1), these data represent a freeze frame of the effects of NF-YA depletion on muscle homeostasis, with the main consequence being the depletion of quiescent and activated cells. The results could be indicative of i) a quiescent-to-differentiation transition of SCs that do not undergo DNA replication or ii) a very rapid DNA replication process before commitment of SCs towards early differentiation. To discriminate between these two hypotheses, we followed SCs from quiescence to differentiation with a time course analysis of EDL myofibers cultured in suspension (new Figures 4H and 4I).

All these new data have been added to the revised manuscript.

Rescue experiments have been done as described above, but not included in the manuscript (see answer to point 1).

5. *Follow up on 4, but inject live mice with EDU and then stain TA cross sections. If SC are breaking quiescence then they should be incorporating more EDU in the KO.*

We performed the experiment accordingly to Referee suggestion. The results, shown in new Figure 4F, demonstrated a decrease in the percentage of Pax7+/EdU+ cells during normal muscle homeostasis after 7 days from the last tamoxifen administration. This represent the final consequence of NF-YA depletion, as demonstrated by time course experiments in isolated myofibers showing that, once activated (by isolation procedure), these cells can incorporate EdU and proceed towards differentiation (new Figure 4I).

6. *Follow up on 4. If the SC are dying then in the RNA-seq data show that apoptotic genes are upregulated. Or find some genes that would explain the loss of SC.*

Cell death was observed in the next few days post-isolation of SCs. The experimental procedure foresees to change culture medium following 24h from cells isolation. This presumably removes the majority of dead cells and consequently reduces the number of both DAPI-stained and Pax7-stained cells in NF-YA^{CKO} culture compared to control ones, although equal number of isolated SCs from NF-YA^{CKO} and NF-YA^{fl/fl} muscles were plated. Despite this, when we extended RNA-seq analysis to larger pathway gene sets (>500 genes), we identified apoptotic GO terms within up-regulated categories so far (new Figure 7A). By means of RT-qPCRs, we validated the transcriptional activation of selected genes (new Figure 7C).

7. *Redo the differentiation assay with more confluent cells and for a longer period of time to see if there's just a delayed activation. (because based on the figure pictures the KO were not nearly confluent enough to differentiate).*

We performed the suggested experiments. Even if the cells were cultured in growth conditions for a longer period compared to control ones (new Figure 5F) or where plated at higher density (new Figure 5G), they were not able to efficiently differentiate. These results have been included in the revised manuscript.

8. *In Figure 1, the authors have done western blot on whole muscle and draw conclusion for the expression in sat cells based on whole muscle analysis. This conclusion is invalid as muscle is a heterogenous tissue and contain numerous non-myogenic cells.*

Western blot analysis of total extracts from control and CTX-injured muscles allowed us to detect NF-YA expression only following injury (Figures 1C and D). As pointed out by the Referee, various myogenic and non-myogenic cells could account for increased NF-YA levels. Indeed, this experiment did not lead us to conclude that NF-YA was expressed in SCs, but other experiments were done and showed in Figure 1 to draw this conclusion. Firstly, we identified NF-YA expression in the nuclei of centrally nucleated fibers following CTX administration (Figure 1D). This suggested us that NF-YA could be expressed in muscle SCs activated following injury. We therefore analyzed by Western blot different cell populations that inhabit the injured muscle tissue, among which SCs, FAPs (Fibro Adipogenic Progenitors) and macrophages (Figure 1E). This result unequivocally showed that NF-YA, and specifically NF-YAL, is expressed in SCs, but also in other cell types, as expected.

9. *Figure 2B: Claims that SC are dying or breaking quiescence yet show even higher pax7 expression in KO under normal conditions, wouldn't that imply that they are actually more quiescent? Also the eMyHC (same with myogenin, myf5 does look real to me) is not convincing to me, they shouldn't be comparing it to steady state, it should be control vs KO in both states. Also, they are using S.E.M and not Standard deviation. I understand that if they used the WT as 1 in both conditions it would be misleading, but seeing 1/4 reduction is not very convincing and it only looks cool when you go 200 vs 150, instead of 1 vs 0.75*

Thanks to this comment, we further analyzed the proportion of quiescent, activated and differentiating cells in NF-YA^{cKO} and NF-YA^{fl/fl} muscles (both in TA sections and in isolated myofibers, see response to point 4 and new Figures 4E and H). The results highlighted that if SCs survive to NF-YA loss, they exit the quiescence and proceed towards proliferation and differentiation. Indeed, percentage of quiescent cells is reduced both in TA sections and in isolated myofibers analyzed at time t 0h (new Figures 4E and H). Despite this, we observed increased levels of Pax7 expression, which is not associated to increased protein levels. As above described, we speculate that additional post-transcriptional mechanisms can participate in the regulation of Pax7 expression (see point 2, Referee 1). This would be an interesting point to be developed.

We calculated S.D. in place of S.E.M. and we changed all the histograms, accordingly. As for RT-qPCR graphs (Figure 2B), we agree with the Referee that representing data as relative expression with respect to NF-YA^{fl/fl} as 1 in both conditions would be misleading. The aim of this analysis was to determine whether SCs are able to activate the transcription of key myogenic genes following muscle damage in the two mouse models, NF-YA^{cKO} and NF-YA^{fl/fl}. Therefore, we decided to show the transcript levels of CTX-muscle as expression relative to untreated-muscle, set at 1 in both models (CTX-NF-YA^{cKO}/ Ctr-NF-YA^{cKO} and CTX-NF-YA^{fl/fl}/ Ctr-NF-YA^{fl/fl}) (as described in Ref. 11).

10. *Figure 2C: through staining that there is a marked expression of eMyHC at 5 days (and say that WT has a more defined staining but I see no difference between, but in WB there is clearly less in the KO.)*

We believe that immunofluorescence assay can well show whether eMyHC is expressed in the area involved in CTX-induced regeneration, but does not allow accurate quantification of its expression in the whole muscle. Western blot has established itself as a reliable technique for gathering quantitative data. We modified the text describing Figure 2A, as suggested by the Referee.

11. Figure 2D: really bad western blot.

We changed Western blot, accordingly. The uncropped western blot is available in Source data file.

12. Figure 3A: Those fiber figure is low quality. And I fail to see the point of it, in all conditions NF-YA is present at all times, it hardly screams that it has a key function important for quiescence and differentiation if it is always present. They claim in the later panels that NF-YA is greatly reduced in differentiation, but the staining really doesn't show that.

We changed immunofluorescence images, accordingly to the Referee comment.

As stated before, we are fully convinced that the most accurate way to quantify protein expression levels is through western blot assay. Moreover, anti-NF-YA antibodies work very well in western blot and have been validated in a multitude of published papers.

Immunofluorescence on isolated myofibers allowed us to demonstrate that NF-YA is expressed in quiescent, proliferating and differentiating SCs. We did not perform a quantification of immunofluorescence images; therefore, we did not draw any conclusion on the possible modulation of NF-YA expression levels when the cells pass from proliferation to differentiation. We came to this conclusion following western blot analysis showed in Figure 3C that has been replaced with a time course analysis.

13. Figure 4B: Change the y-axis, # of pax7+ cells/fiber is wrong for cross sections. Just do # of Pax7+/mm2.

As described in the text, Figure 4B (now Figure 4G) represents the number of Pax7+ cells retained in isolated EDL myofibers. The quantification of Pax7+ cells in TA sections has been now added (new Figure 4A, right panel) and represented as Pax7+ cells/mm².

14. Figure 4E: the panel here is misleading. They don't say at what time post isolation this was done and it would be better represented as a percentage, not a total number as they establish in the previous panels that the KO has less cells to begin with. Also, I would like to see a total number of SC per fiber as confirmation for previous panels.

We performed additional experiments to better follow the fate of SCs in isolated myofibers. Figures 4E and 4F have been replaced (see new Figures 4G-I).

15. Figure 4F: Same as figure 4E needs to be percentage not total number as KO has less cells. Also the staining pictures are very low quality and need to be their own panel in my mind.

We replaced Figure 4F with new Figure 4H, which shows the relative proportion of Myog+/MyoD+, Myog-/MyoD+ and MyoG+/MyoD- cells following 72 hours from myofibers isolation.

16. Figure 5 A,B and C don't line up. A and B are claiming that % of Pax7 cells are lower in KO but C is saying the complete opposite that it's actually higher in the KO.

Since the staining of Pax7 and MyoD was performed in growth culture condition, we decided to represent the relative distribution of quiescent Pax7+/MyoD- and activated Pax7+/MyoD+ within total Pax7+ cells (see new Figure 5B). Differentiating cells were analyzed following the switch to differentiation medium (see new Figures 5F and G).

17. Figure 5 D is strange data, they only added EDU for 2 hours, there's no way they got 40% positive cells in WT in such short time. Also the time EDU was added is not stated in the figure legend but it is in the materials and methods.

We followed the experimental procedure published in other research articles^{12,13}, where they similarly cultured SCs with EdU for 90 min or 2 h, respectively. They identified 30% and 40% of EdU⁺/PAX7⁺ cells in control cells. Therefore, we are not surprised that after 2 hours of EdU incubation in growth medium, about 20% of EdU⁺/Pax7⁺ cells were detected in NF-YA^{fl/fl} control cells and rised to 26% in NF-YA^{cKO} cells. All the details related to experimental procedure were described in Methods section, but also added in the Figure legend.

18. Figure 5 F is pointless in my mind. It's just the % of myogenin cells in normal growth culture.

We removed this Figure as suggested by the Referee and we performed new experiments to analyze cell differentiation (new Figures 5F and G).

19. Figure 5 G: It would be nice if they left it in DM for longer to see if eventually the KO will differentiate properly.

We performed the suggested experiment and we observed that, even if incubated in differentiation medium for longer time or plated at higher density to reach the same density of control cells, NF-YA^{cKO} cells did not differentiate properly (new Figures 5F and G).

20. Figure 7B/D: that break in the y axis is completely unnecessary and is only there to make it look like there's a bigger difference than there is.

We are sorry for this inaccuracy. We changed the axis, accordingly.

Yours sincerely,
Carol Imbriano

References

1. Olguín, H. C. & Pisconti, A. Marking the tempo for myogenesis: Pax7 and the regulation of muscle stem cell fate decisions. *Journal of Cellular and Molecular Medicine* **16**, 1013–1025 (2012).
2. Bustos, F. et al. NEDD4 Regulates PAX7 Levels Promoting Activation of the Differentiation Program in Skeletal Muscle Precursors. *Stem Cells* **33**, 3138–3151 (2015).
3. González, N. et al. CK2-dependent phosphorylation is required to maintain Pax7 protein levels in proliferating muscle progenitors. *PLoS One* **11**, (2016).
4. Benatti, P. et al. Specific inhibition of NF-Y subunits triggers different cell proliferation defects. *Nucleic Acids Res.* **39**, 5356–68 (2011).
5. Puri, P. L. et al. A myogenic differentiation checkpoint activated by genotoxic stress. *Nat. Genet.* **32**, 585–593 (2002).
6. Latella, L. et al. DNA damage signaling mediates the functional antagonism between replicative senescence and terminal muscle differentiation. *Genes Dev.* **31**, 648–659 (2017).
7. Simonatto, M. et al. Coordination of cell cycle, DNA repair and muscle gene expression in myoblasts exposed to genotoxic stress. *Cell Cycle* **10**, 2355–2363 (2011).
8. Gurtner, A. et al. Requirement for down-regulation of the CCAAT-binding activity of the NF-Y transcription factor during skeletal muscle differentiation. *Mol. Biol. Cell* **14**, 2706–2715 (2003).
9. Basile, V. et al. NF-YA splice variants have different roles on muscle differentiation. *Biochim. Biophys. Acta - Gene Regul. Mech.* **1859**, 627–638 (2016).
10. Libetti, D. et al. The Switch from NF-YA1 to NF-YAs Isoform Impairs Myotubes Formation. *Cells* **9**, 789 (2020).

11. *Chen, H. H. et al. NRIP is newly identified as a Z-disc protein, activating calmodulin signaling for skeletal muscle contraction and regeneration. J. Cell Sci. 128, 4196–4209 (2015).*
12. *Ogura, Y. et al. TAK1 modulates satellite stem cell homeostasis and skeletal muscle repair. Nat. Commun. 6, 1–17 (2015).*
13. *Su, Y. et al. Fate decision of satellite cell differentiation and self-renewal by miR-31-IL34 axis. Cell Death Differ. 27, 949–965 (2020).*

Reviewers' Comments:

Reviewer #1:

Remarks to the Author:

Thank you for your thoughtful and thorough consideration of the comments provided. The manuscript is substantially improved. Congratulations on a very nice paper.

Reviewer #2:

Remarks to the Author:

The authors have performed a number of new experiments for revisions and although these new experiments provide some improvement to the paper, there are still many major concerns throughout the paper.

Major concerns:

1. The cKO mice have an average knockdown of NF-YA around 40% at the level of mRNA. This level of mRNA knockdown clearly shows that knockout experiment has not worked optimally.
2. Figure 2 displays how eMyHC expression changes throughout regeneration. At 5 days, in the IF there is clearly eMyHC staining in the cKO condition and the qPCR shows the expression of eMyHC, albeit at a reduced level compared to the WT. However, the WB for the 5 day injured muscle shows no eMyHC in the cKO, and the loading controls are uneven. Further, the 15 day injured muscle WB has a very inconsistent level of eMyHC.
3. Figure 3C is a WB of myogenic factors, MyHC and NF-YA throughout differentiation. However, the data presented goes against previously published data and well established consensus on the expression of these myogenic factor in differentiation. The authors show that myogenin is only present at 1 day differentiation, and MyoD is only detectable in growth media. Neither of which is supported by previous publications, where myogenin is detectable for 72 hours in differentiation media and MyoD is also highly expressed during differentiation as well.
4. Figure 4H claims that at 0 hr post isolation nearly 50% of all EDL associated satellite cells are MyoD positive. This is much higher than established percentage in the field, which would be nearer to 1%. (Goel, A. J., M. K. Rieder, H. H. Arnold, G. L. Radice and R. S. Krauss (2017). "Niche Cadherins Control the Quiescence-to-Activation Transition in Muscle Stem Cells." *Cell Rep* 21(8): 2236-2250. Figure 4B). Figure 4E also claims that approximately 30% of MuSCs in an uninjured TA muscle are expressing MyoD.
5. Figure 4H and I are inconsistent in the percentage of cells expressing Pax7 at 24 hours. In Figure 4I at 24 hours it is shown that approximately 90% and 75% of cells are Pax7 positive in WT and cKO respectively, yet in Figure 4H the number of Pax7 positive cells at 24 hours post isolation is less than 50% in both conditions.

6. Figure 4F, 4I and 5C looked at EDU incorporation both in vivo and in vitro. To begin with, the data between these figures are also inconsistent. When injected in vivo the number of EDU positive MuSCs is lower in the cKO mice, however when performed on isolated myofibers, the cKO mice have 100% of their cells that have incorporated EDU, and in vitro the cKO displays greater EDU incorporation than the WT. How do the authors explain these discrepancies between their experiments? Further, the level of EDU incorporation seen in Figure 4F, vastly exceeds what is typically seen in in vivo EDU experiments, where 24 hours after injection the percentage of EDU positive MuSC is near 1%, not 8%. (de Morree, A., C. T. J. van Velthoven, Q. Gan, J. S. Salvi, J. D. Klein, I. Akimenko, M. Quarta, S. Biressi and T. A. Rando (2017). "Staufen1 inhibits MyoD translation to actively maintain muscle stem cell quiescence." *Proc Natl Acad Sci U S A* 114(43): E8996-E9005. Figure 4B)

7. Figure 4G shows that the WT myofibers have on average 2 Pax7+ cells per fiber at 0 hours post isolation, which is significantly lower than what is seen by other groups in previous publications. (Goel, A. J., M. K. Rieder, H. H. Arnold, G. L. Radice and R. S. Krauss (2017). "Niche Cadherins Control the Quiescence-to-Activation Transition in Muscle Stem Cells." *Cell Rep* 21(8): 2236-2250. Figure 1G) (Kuang, S., K. Kuroda, F. Le Grand and M. A. Rudnicki (2007). "Asymmetric self-renewal and commitment of satellite stem cells in muscle." *Cell* 129(5): 999-1010. Figure 1C)

Minor Concerns:

The figures are often unclear on the exact data that is being shown and the reader often needs to consult the figure captions for clarification. It would help if the authors added more clear labelling to their panels. For example, Figure 1 B and C would be clearer if it specified that these experiments were performed on whole muscle and that the injury condition was 5 days post injury. Same in Figure 2C and D, where the only difference between the two panels was the length of recovery in the injured condition, yet that is not made clear in the figure itself.

Figure 1D claims that the presence of NF-YA positive myonuclei is most likely derived from MuSCs that fused with the fiber during regeneration. This is purely speculative, to further prove this point it would be best to isolate regenerating EDL myofibers and stain for NF-YA, if all of the centrally located nuclei in a fiber are positive, then that would indicate that the upregulation of NF-YA during injury may instead be a response from the myofibers and not necessarily from the MuSCs, this would also help explain the large increase in NF-YA seen in the injured muscle as MuSCs are not the principal component of skeletal muscle, but myofibers are. However if only a small subset of the myonuclei are positive for NF-YA, that would be much stronger evidence for the hypothesis that it is coming from freshly fused MuSCs.

Figure 3A shows EDL associated MuSCs at different timepoints, however the quiescent condition (0 hours post isolation) appears to be closer to 24 hours post isolation as it is very rare to have 4 MuSC adjacent to one another and they also appear to be doublets. Further the NF-YA, in quiescent MuSCs appears to be primarily cytoplasmic, meaning it would have no real function as it is a transcription factor. The proliferation condition (24 hours post isolation) appears to be more consistent with what would be seen at 48 hours post isolation. The differentiation condition has almost no cells that are actually myogenin positive (in the nuclei), it would be best to get a different representative picture for that condition.

Figure 3C is labeled a GM, t1, t3 and t5. I assume the authors meant GM, DM1, DM3, DM5.

Figure 3D is labelled relative mRNA levels of 3 genes between proliferating and differentiating MuSCs. But what is it relative to? None of the genes or conditions have an average expression of 1.

Figure 4D should have a representative staining picture associated with it.

Figure 4L, when EDU is at 100% it becomes less informative. It would be better to optimize the timing of the EDU dose to glean more information. It is possible that the cKO nears 100% EDU incorporation when the WT is still at 50%, but as time continues the WT will still be incorporating EDU while the cKO is capped at 100%. Same with the myogenin section, also there is a Myog+/EDU- category in the legend that is unused and should be removed.

Figure 5G immunofluorescence would look better with a brightfield channel to more clearly see the size and shape of the myotubes.

Figure 6 C-F, a more appropriate control would be an IgG affinity matched control instead of % input.

Figure 7 E-G, although the authors clearly show that when NF-YA1 is reintroduced to the knockout cells there is a reduction in some of the components of the DNA repair mechanism, there is no reduction in γ H2AX, which the authors use as their marker for DNA damage.

It would be a good idea to maintain the induced knockout mice on tamoxifen chow after the IP injections.

"The difference in CSA and percentage of centrally nucleated myofibers between NF-YAfl/fl and NF-YA^{CKO} mice was still evident following 60 days after injury and 120 was associated with reduced muscle weight (Fig. 2G-I and Suppl. Fig. 1F)" The figures referenced here are improperly labeled, Fig. 2G-I does not exist, it is Fig. 1G & H.

Supplemental Figure 1D, in text it is described as freshly isolated myofibers, yet there are at least 8 MuSCs that can be seen in the representative picture of a single myofiber, which again is extremely unlikely to be seen in a 0 hour post isolation myofiber. Further, it also goes against the count of Pax7+ cells/myofiber seen in figure 4G that claimed only on average 2 per fiber.

The MuSCs associated with EDL myofibers in the cKO appear to be undergoing rapid

differentiation, having a higher proportion of MuSCs expressing myogenin at 48 hours, yet at 72 hours these same cells appear to have difficulty transitioning from a MyoD+/Myog+ state to a more differentiated MyoD-/Myog+ state. It would be interesting if the authors could more clearly explain this phenomenon. It is possible that the cKO MuSCs are being driven towards differentiation at the expense of proliferation, yet are also somehow unable to fully enter the differentiation program. I would like it if the authors stained their in vitro differentiating cells with MyoD and Myogenin, to see whether the results seen on the EDL can be replicated.

Reviewer #1 (Remarks to the Author):

The manuscript is substantially improved. Congratulations on a very nice paper. We sincerely thank the Reviewer for her/his appreciation for our work.

Reviewer #2 (Remarks to the Author):

The authors have performed a number of new experiments for revisions and although these new experiments provide some improvement to the paper, there are still many major concerns throughout the paper.

We are very grateful to the Reviewer for her/his appropriate suggestions and for proposed corrections to improve our revised manuscript. We have addressed all the new issues and we hope that our point-to-point response and modifications of the manuscript can dispel the Reviewer's doubts. We add a schematic representation of the main findings from our study to highlight the effects induced by reduced NF-Y activity in muscle homeostasis and regeneration (Figure 8).

Major concerns:

1. The cKO mice have an average knockdown of NF-YA around 40% at the level of mRNA. This level of mRNA knockdown clearly shows that knockout experiment has not worked optimally.

This result was already shown in the first version of our manuscript, nevertheless it was not highlighted as a critical point by none of the Reviewers. Despite this, we would like to share our considerations with the Reviewer. The complete KO of NF-YA is embryonically lethal¹ and it is not possible to study NF-Y in skeletal muscle biology through this model. The specific aim of our research was to investigate the role of NF-Y in adult skeletal muscle stem cells, therefore we created a conditional knock out mouse model. Other groups generated NF-YA conditional KO mice to study NF-Y in adult tissues. The group of S.N. Maity knocked out NF-YA postnatally exclusively in liver by crossing Alb-Cre mice with NF-YA^{flox/flox}. Quantitative PCR was done (but not shown) to measure deleted and wt alleles in tissue DNAs and approximately 60% of the NF-YA^{flox} allele was deleted in the liver of NF-YA^{flox/flox}/Alb-Cre mice at 4 weeks of age², but NF-YA mRNA levels were not investigated in Maity's paper, published in Cancer Research journal. Conditional deletion of NF-YA in postmitotic mouse neurons was done and published in Nature Communication journal by the group of Nobuyuki Nukina³. Quantitative RT-PCR analysis of NF-YA transcript in cortex and hippocampus of 5-week-old NF-YA mice showed a decrease of about 40% (see Figure 2e and Suppl. Figure 4b).

Our results, obtained by using a conditional and also inducible mouse model (NF-YA^{fl/fl};Pax7-Cre mice), are in line with the results above described. We are aware that this cannot be considered a complete knock out, indeed we never used this definition throughout the manuscript. Despite the lack of a complete KO, we observed evident effects on muscle stem cell biology demonstrating that NF-YA reduction is sufficient to perturb muscle homeostasis and regeneration. We hope that the Referee will agree with us that our mouse model allowed to get new insights on the role of NF-Y activity in adult muscle stem cells.

2. Figure 2 displays how eMyHC expression changes throughout regeneration. At 5 days, in the IF there is clearly eMyHC staining in the cKO condition and the qPCR shows the expression of eMyHC, albeit at a reduced level compared to the WT. However, the WB for the 5 day injured muscle shows no eMyHC in the cKO, and the loading controls are uneven. Further, the 15 day injured muscle WB has a very inconsistent level of eMyHC.

As we previously tried to clarify, the immunofluorescence assay allows to identify whether eMyHC is expressed in the area involved in CTX-induced regeneration, but does not accurately quantify its expression in the whole muscle. Quantification of eMyHC in total muscle depends on the number and dimension of eMyHC-positive fibers, as well as on the expression levels of eMyHC in each myofiber. Western blot on total muscles following 5 days from injury shows high levels of eMyHC

staining in NF-YA^{fl/fl} mice, evident also in NF-YA^{cKO} mice, despite at lower levels. We agree with the Referee that loading control are uneven, therefore we performed a new western blot analysis. The image of Figure 2C has been replaced.

The immunofluorescence of Figure 2A shows the presence of multiple fibers still highly positive for eMyHC in 15 days NF-YA^{cKO} muscles, while weak eMyHC positivity is detected in NF-YA^{fl/fl} mice. We replaced the selected enlargement of NF-YA^{fl/fl} muscle to better highlight these differences. Western blot of Figure 2D shows eMyHC levels in whole extracts of 15 days post-injury muscles. Consistently with the immunofluorescence, higher eMyHC levels are detected in NF-YA^{cKO} compared to NF-YA^{fl/fl} muscles, suggesting a delay in the regenerative response when NF-YA expression is reduced in satellite cells. We added information on CTX-days on each figure panel to avoid misunderstanding, as suggested by the Referee (see minor concerns), and we modified the text in the manuscript to make the result clearer.

3. Figure 3C is a WB of myogenic factors, MyHC and NF-YA throughout differentiation. However, the data presented goes against previously published data and well established consensus on the expression of these myogenic factor in differentiation. The authors show that myogenin is only present at 1 day differentiation, and MyoD is only detectable in growth media. Neither of which is supported by previous publications, where myogenin is detectable for 72 hours in differentiation media and MyoD is also highly expressed during differentiation as well.

We performed western blot on different cell extracts from a new experiment. We completely replaced western blot panel, accordingly. Myogenin, which we previously showed to be expressed at DM1 and to a lesser extent at DM3, was better detected in new cell extracts loaded in higher quantity. As MyoD concerns, we probed the membrane with a new rabbit polyclonal antibody from Proteintech (18943-1-AP) that works well in western blot (<https://www.ptglab.com/products/MYOD1-Antibody-18943-1-AP.htm>). We used Proteintech antibody in place of the rabbit anti-MyoD (C-20) from Santa Cruz Biotechnology (sc-304), which has been discontinued.

Our results are consistent with other literature data, such as those published by Kuang in Nature Communication journal⁴ that showed a peak of MyoD levels at GM followed by an evident decrease at DM1, and high Myogenin levels at DM1 followed by decreased expression at DM3 and DM5.

4. Figure 4H claims that at 0 hr post isolation nearly 50% of all EDL associated satellite cells are MyoD positive. This is much higher than established percentage in the field, which would be nearer to 1% (Goel, A. J., M. K. Rieder, H. H. Arnold, G. L. Radice and R. S. Krauss (2017). "Niche Cadherins Control the Quiescence-to-Activation Transition in Muscle Stem Cells." Cell Rep 21(8): 2236-2250. Figure 4B).

Our data showed about 40% of Pax7+/MyoD+ cells after myofibers isolation from NF-YA^{cKO} and NF-YA^{flox/flox} muscles in physiological conditions. We are aware that the percentage of Pax7+/MyoD+ cells at time 0h after isolation is higher than what is shown in other relevant published papers (e.g. Rando and Bonaldo described in their Nature Communication paper about 20% of MyoD+ cells [Pax7+/MyoD+ and Pax7-/MyoD+] in t 0h myofibers⁵). It has been well described the rapid activation response of muscle stem cells that occurs after myofibers isolation, as well as after satellite cells isolation⁶⁻⁸. Therefore, the percentage of MyoD+ cells in myofibers fixed after isolation strictly depends on the time required for the procedure of isolation itself, as the simple process of isolation can initiate the activation process. We followed the detailed procedure described by M.A. Rudnicky⁹ and our results fit with the percentage of Pax7+/MyoD+ cells detected between 1h (21%) and 4h (72%) from isolation. Although our aim was not to determine the absolute quantification of myogenic populations in NF-YA^{cKO} myofibers, rather to compare the activation and differentiation process of SCs between NF-YA^{cKO} and NF-YA^{flox/flox} muscles, we agree that it could be misleading the description of the first time point as t0 (time 0h). We therefore

replaced time t0 with d0 (day 0), as described in other papers¹⁰, to better identify the first time point we were able to obtain following myofibers isolation (see materials and methods, manuscript and figure legend).

Figure 4E also claims that approximately 30% of MuSCs in an uninjured TA muscle are expressing MyoD.

To address the concerns raised in the previous round of review by the Referee, who suggested to analyze the myogenic populations in TA uninjured muscle sections, we used the contralateral leg of the same animals that received CTX administration (Figure 4E). This is a common procedure used in various published studies^{11–13}, in particular for molecular analysis¹⁴, which allows to minimize the number of animals used per experiment according to the principles of the 3Rs (Replacement, Reduction and Refinement), embedded in national and international legislation and regulations on the use of animals in scientific procedures. As observed by the Referee, our experiments showed a higher number of activated Pax7+/MyoD+ cells (Figure 4E), as well as a higher percentage of EdU+ cells (see response to point 6), in contralateral muscles with respect to published data from control muscles. We suggest that this control sample could not properly overlap the muscle homeostatic condition. Indeed, it has been described the existence of a G_{Alert} status in SCs distant from the site of an injury, characterized by increased propensity to cycle *in vivo* and reduced time to first division¹⁵. We are conscious that this might be misleading, therefore we decided to perform new stainings on TA sections of NF-YA^{flox/flox} and NF-YA^{cKO} mice in physiological conditions. Differently from what previously observed, we detected fewer Pax7+/MyoD+ cells in NF-YA^{flox/flox} control muscles. NF-YA^{cKO} showed a significant increase of Pax7+/MyoD+ population. Similarly to what stated in the previous revised version of our manuscript, these results demonstrate that NF-YA^{cKO} SCs are more prone to exit from quiescence during muscle homeostasis (see new Figure 4E). Obvious questions arise from our previous data on the contralateral leg on the possible functional role of NF-YA in the G_{Alert} status and the signaling pathways activated following NF-YA reduction in SCs that we hope to address in the near future.

5. Figure 4H and I are inconsistent in the percentage of cells expressing Pax7 at 24 hours. In Figure 4I at 24 hours it is shown that approximately 90% and 75% of cells are Pax7 positive in WT and cKO respectively, yet in Figure 4H the number of Pax7 positive cells at 24 hours post isolation is less than 50% in both conditions.

Figure 4H and 4I similarly show the percentage of myogenic populations following 24 hours from isolation, but the cells were maintained in two different culture conditions and therefore cannot be directly compared. While Figure 4H shows myofibers that were cultured in growth medium as soon as isolated and maintained for 24h in the same medium, the experiment shown in Figure 4I was performed by maintaining myofibers for 20 hours in growth medium supplemented with EdU 10µm and then, after two fibers washes, incubated for additional 4 hours in fresh growth medium. Since the aim of EdU administration to myofibers was to determine whether NF-YA^{cKO} SCs were able to escape DNA replication and directly differentiate (please, see minor concern), we decided to focus the attention only on the distribution of cells at 72h (3 days) time point. Moreover, we removed the MyoG+/EdU- category in the legend, as suggested by the Referee (minor concerns).

6. Figure 4F, 4I and 5C looked at EDU incorporation both in vivo and in vitro. To begin with, the data between these figures are also inconsistent. When injected in vivo the number of EDU positive MuSCs is lower in the cKO mice, however when performed on isolated myofibers, the cKO mice have 100% of their cells that have incorporated EDU, and in vitro the cKO displays greater EDU incorporation than the WT. How do the authors explain these discrepancies between their experiments? Further, the level of EDU incorporation seen in Figure 4F, vastly exceeds what is typically seen in in vivo EDU experiments, where 24 hours after injection the percentage of EDU positive MuSC is near 1%, not 8%. (de Morree, A., C. T. J. van Velthoven, Q. Gan, J. S. Salvi, J. D. Klein, I. Akimenko, M. Quarta, S. Biressi and T. A. Rando (2017). "Staufen1 inhibits MyoD

translation to actively maintain muscle stem cell quiescence." Proc Natl Acad Sci U S A 114(43): E8996-E9005. Figure 4B).

EdU was administered to mice following 7 days from the last TMX administration, but similarly to what described above (see point 4), we used the contralateral leg of mice that received CTX as uninjured condition. The high number of EdU+ cells in NF-YA^{flox/flox} could be the consequence of a partial activation of SCs distant from the site of an injury (see point 4). Taking into consideration the legitimate concern raised by the Referee, we performed new *in vivo* experiments and we analyzed EdU incorporation in muscle of mice in physiological condition or following CTX-injury to enhance SCs activation. Our new results (Figure 4F) allowed to identify few EdU+ cells in uninjured muscles of NF-YA^{flox/flox} mice (about 2%), which significantly increased in NF-YA^{CKO} ones. Similarly, in CTX-injured muscle, NF-YA^{CKO} mice showed a higher percentage of Pax7+ cells positive for EdU+ when compared to NF-YA^{flox/flox} mice (Suppl. Figure 1H).

7. *Figure 4G shows that the WT myofibers have on average 2 Pax7+ cells per fiber at 0 hours post isolation, which is significantly lower than what is seen by other groups in previous publications. (Goel, A. J., M. K. Rieder, H. H. Arnold, G. L. Radice and R. S. Krauss (2017). "Niche Cadherins Control the Quiescence-to-Activation Transition in Muscle Stem Cells." Cell Rep21(8): 2236-2250. Figure 1G) (Kuang, S., K. Kuroda, F. Le Grand and M. A. Rudnicki (2007). "Asymmetric self-renewal and commitment of satellite stem cells in muscle." Cell 129(5): 999-1010. Figure 1C)*

We improperly named the y axis, which refers to Pax7+ cells/EDL myofiber nuclei (%) and not to Pax7+ cells/EDL myofiber. We deeply apologize for our inaccuracy. We replaced y axis title. This representation of the results, as reported by other groups (Fig. 4A - Ref.¹⁶, Fig. 6B - Ref.¹⁷) allows to evaluate Pax7+ cells in each myofiber (frequency), independently from myofiber length and myonuclei number. Taking into consideration the number of myonuclei per myofiber in 6/8 weeks old mice (about 250/300 nuclei/myofiber)¹⁸, our results (2.3 +/-0.5 Pax7+ cells/100 nuclei) are consistent with what reported in literature (about 4 to 6 Pax7+ cells per myofiber)¹⁸⁻²⁰.

Minor Concerns:

The figures are often unclear on the exact data that is being shown and the reader often needs to consult the figure captions for clarification. It would help if the authors added more clear labelling to their panels. For example, Figure 1 B and C would be clearer if it specified that these experiments were performed on whole muscle and that the injury condition was 5 days post injury. Same in Figure 2C and D, where the only difference between the two panels was the length of recovery in the injured condition, yet that is not made clear in the figure itself.

We thank the Reviewer for the suggestion. We added these details, accordingly.

Figure 1D claims that the presence of NF-YA positive myonuclei is most likely derived from MuSCs that fused with the fiber during regeneration. This is purely speculative, to further prove this point it would be best to isolate regenerating EDL myofibers and stain for NF-YA, if all of the centrally located nuclei in a fiber are positive, then that would indicate that the upregulation of NF-YA during injury may instead be a response from the myofibers and not necessarily from the MuSCs, this would also help explain the large increase in NF-YA1 seen in the injured muscle as MuSCs are not the principal component of skeletal muscle, but myofibers are. However if only a small subset of the myonuclei are positive for NF-YA, that would be much stronger evidence for the hypothesis that it is coming from freshly fused MuSCs.

We are sorry if our interpretation of Figure 2D was not clear. The immunofluorescence showed NF-YA expression in some centrally nucleated myofibers, in addition to other interstitial cells. We replaced the immunofluorescence image with a new one that better allows the identification of different cellular types positive for NF-YA nuclear staining, such as interstitial cells and some centrally nucleated fibers. It is unquestionable that NF-YA is expressed in various cellular components of skeletal muscle following injury (such as FAPs and macrophages), as demonstrated

by western blot shown in Figure 1E, which could help to explain the large increase in NF-YA1 seen in injured muscle (Figure 1C). From the immunofluorescence of Figure 1D, we did not draw the conclusion that NF-YA is expressed in SCs in regenerating muscles, but we hypothesized that “NF-YA could be expressed in muscle SCs activated following injury”. We have reworded the text to avoid misunderstanding.

Figure 3A shows EDL associated MuSCs at different timepoints, however the quiescent condition (0 hours post isolation) appears to be closer to 24 hours post isolation as it is very rare to have 4 MuSC adjacent to one another and they also appear to be doublets. Further the NF-YA, in quiescent MuSCs appears to be primarily cytoplasmic, meaning it would have no real function as it is a transcription factor. The proliferation condition (24 hours post isolation) appears to be more consistent with what would be seen at 48 hours post isolation. The differentiation condition has almost no cells that are actually myogenin positive (in the nuclei), it would be best to get a different representative picture for that condition.

We replaced the images, accordingly.

Figure 3C is labeled a GM, t1, t3 and t5. I assume the authors meant GM, DM1, DM3, DM5.

We changed Figure caption, accordingly.

Figure 3D is labelled relative mRNA levels of 3 genes between proliferating and differentiating MuSCs. But what is it relative to? None of the genes or conditions have an average expression of 1. RT-qPCR data were represented as ΔC_P values that determine target mRNA expression levels relatively to the reference gene. We agree that it would be clearer to quantify transcript levels as mRNA expression of differentiating cells relatively to proliferating ones (arbitrarily set at 1). We changed the histograms, accordingly.

Figure 4D should have a representative staining picture associated with it.

We included a representative staining picture, accordingly.

Figure 4L, when EDU is at 100% it becomes less informative. It would be better to optimize the timing of the EDU dose to glean more information. It is possible that the cKO nears 100% EDU incorporation when the WT is still at 50%, but as time continues the WT will still be incorporating EDU while the cKO is capped at 100%. Same with the myogenin section, also there is a Myog+/EDU- category in the legend that is unused and should be removed.

We guess the concern refers to Figure 4I. The aim of the experiment was to determine whether SCs from NF-YA^{cKO} myofibers were committed towards differentiation bypassing the S-phase. Therefore, we were interested in identifying whether there was a population that did not incorporate EdU among Myogenin+ cells. For this reason, we used the experimental condition in which EdU was administered in excess and for enough time to allow all replicating cells to be labeled. The presence of MyoG+/EdU- would mean that some primary myogenic cells experienced a transition from quiescence to differentiation without undergoing DNA replication. We decided to show the results only in differentiating conditions (day 3) to avoid misinterpretation of the experiment. The MyoG+/EdU- category has been removed, according to Referee suggestion.

Figure 5G immunofluorescence would look better with a brightfield channel to more clearly see the size and shape of the myotubes.

We are sorry but we did not take snapshots of the bright field channel, as we were interested in MyHC positivity in cell cultures.

Figure 6 C-F, a more appropriate control would be an IgG affinity matched control instead of % input.

Many respected groups expert in chromatin techniques showed their results by normalizing data as % of input value. This is a shared procedure^{21,22}. If different experimental groups are compared, accurate quantification of input chromatin is fundamental to ensure that identical amounts are used in each IP reaction. Of course, this does not allow to distinguish if the positive target region is being specifically or non-specifically bound by the antibody. This is why the same sample has been used for RT-qPCR with oligonucleotides for Chrm7 satellite region, which should not be amplified unless non-specific binding occurs. In addition to this experimental negative control, we have always performed our ChIP experiments with IgG antibody and we now added these results in the histograms, according to Referee request.

Figure 7 E-G, although the authors clearly show that when NF-YA1 is reintroduced to the knockout cells there is a reduction in some of the components of the DNA repair mechanism, there is no reduction in γ H2AX, which the authors use as their marker for DNA damage.

The Reviewer is right: Western blot of Figure 7G does not show γ H2AX decrease following NF-YA1 overexpression in NF-YA^{cKO} SCs. We performed a new experiment and we decided to use total histone H2A in place of histone H3 as loading control in western blot analysis. Presumably, this represents a more appropriate control for γ H2AX expression. As shown in new Figure 7G, a weak decrease in γ H2AX levels can be observed. The result suggests that re-expression of NF-YA can modulate the transcription of DDR genes, and, despite slightly, the activation of the DDR pathway. To be consistent with new Figure 7G, we replaced Figure 7E with a new western blot in which we run control and NF-YA^{cKO} cellular extracts using H2A and vinculin as loading controls.

It would be a good idea to maintain the induced knockout mice on tamoxifen chow after the IP injections.

We followed the procedure used by many other groups that performed conditional KO in adult SCs through Tamoxifen induction²³⁻²⁷. Intra-peritoneal administration of Tamoxifen was used before CTX injury even when muscles were harvested following 50-60 days^{12,27}. Despite this, to maintain NF-YA genetic knock out up to 60 days, we administered Tamoxifen once a week, as shown by Olson's group in PNAS paper²⁸. We added this detail in the schematic outlining strategy of Tamoxifen and Cardiotoxin treatment of Figure 1F.

It could be that administration of Tamoxifen chow would increase the percentage of NF-YA deletion, but we could not change the procedure because not described in the protocol approved by National Institute of Health (Ministero della Salute) (project n. 404/2015-PR and 699/2019-PR).

“The difference in CSA and percentage of centrally nucleated myofibers between NF-YA^{fl/fl} and NF-YA^{cKO} mice was still evident following 60 days after injury and 120 was associated with reduced muscle weight (Fig. 2G-I and Suppl. Fig. 1F)” The figures referenced here are improperly labeled, Fig. 2G-I does not exist, it is Fig. 1G & H.

We are sorry for the inaccuracy. We changed the text, accordingly.

Supplemental Figure 1D, in text it is described as freshly isolated myofibers, yet there are at least 8 MuSCs that can be seen in the representative picture of a single myofiber, which again is extremely unlikely to be seen in a 0 hour post isolation myofiber. Further, it also goes against the count of Pax7+ cells/myofiber seen in figure 4G that claimed only on average 2 per fiber.

We replaced the images with new ones representative of day 0 condition (see major concerns, point 4).

The MuSCs associated with EDL myofibers in the cKO appear to be undergoing rapid differentiation, having a higher proportion of MuSCs expressing myogenin at 48 hours, yet at 72 hours these same cells appear to have difficulty transitioning from a MyoD+/Myog+ state to a more differentiated MyoD-/Myog+ state. It would be interesting if the authors could more clearly

explain this phenomenon. It is possible that the cKO MuSCs are being driven towards differentiation at the expense of proliferation, yet are also somehow unable to fully enter the differentiation program. I would like it if the authors stained their *in vitro* differentiating cells with MyoD and Myogenin, to see whether the results seen on the EDL can be replicated.

We thank the Reviewer for this suggestion. We stained MyoD+ and MyoG+ cells following 1 and 2 days in differentiating conditions (DM1 and DM2). We detected significant differences in the distribution of myogenic cells at DM2: MyoD+/MyoG+ committed population increased from 49.5 +/- 1.2 in NF-YA^{fl/fl} to 63.5 +/- 2.1 in NF-YA^{cKO} at the expense of more differentiated MyoD-/MyoG+ cells, which decline from 32.7 +/- 2.2 in NF-YA^{fl/fl} to 18.2 +/- 2.6 in NF-YA^{cKO}. These new results have been added in new Figure 5H.

References

1. Bhattacharya, A. *et al.* The B Subunit of the CCAAT Box Binding Transcription Factor Complex (CBF/NF-Y) Is Essential for Early Mouse Development and Cell Proliferation. *Cancer Res.* **63**, (2003).
2. Luo, R., Klumpp, S. A., Finegold, M. J. & Maity, S. N. Inactivation of CBF/NF-Y in postnatal liver causes hepatocellular degeneration, lipid deposition and endoplasmic reticulum stress. *Sci. Rep.* **1**, 136 (2011).
3. Yamanaka, T. *et al.* NF-Y inactivation causes atypical neurodegeneration characterized by ubiquitin and p62 accumulation and endoplasmic reticulum disorganization. *Nat. Commun.* **5**, 3354 (2014).
4. Yue, F. *et al.* Pten is necessary for the quiescence and maintenance of adult muscle stem cells. *Nat. Commun.* **8**, 1–13 (2017).
5. Urciuolo, A. *et al.* Collagen VI regulates satellite cell self-renewal and muscle regeneration. *Nat. Commun.* **4**, 1–13 (2013).
6. Machado, L., Relaix, F. & Mourikis, P. Stress relief: emerging methods to mitigate dissociation-induced artefacts. *Trends in Cell Biology* **0**, (2021).
7. Machado, L. *et al.* In Situ Fixation Redefines Quiescence and Early Activation of Skeletal Muscle Stem Cells. *Cell Rep.* **21**, 1982–1993 (2017).
8. Machado, L. *et al.* Tissue damage induces a conserved stress response that initiates quiescent muscle stem cell activation. *Cell Stem Cell* **28**, 1125–1135.e7 (2021).
9. Brun, C. E., Wang, Y. X. & Rudnicki, M. A. Single EDL myofiber isolation for analyses of quiescent and activated muscle stem Cells. in *Methods in Molecular Biology* **1686**, 149–159 (Humana Press Inc., 2018).
10. Jia, Z. *et al.* A requirement of polo-like Kinase 1 in murine embryonic myogenesis and adult muscle regeneration. *Elife* **8**, (2019).
11. Pesseme, L. *et al.* Regulation of mitochondrial activity controls the duration of skeletal muscle regeneration in response to injury. *Sci. Reports* **9**, 1–11 (2019).
12. Shea, K. L. *et al.* Sprouty1 Regulates Reversible Quiescence of a Self-Renewing Adult Muscle Stem Cell Pool during Regeneration. *Cell Stem Cell* **6**, 117–129 (2010).
13. Zhang, Y. *et al.* Oscillations of Delta-like1 regulate the balance between differentiation and maintenance of muscle stem cells. *Nat. Commun.* **12**, 1–16 (2021).
14. Du, H. *et al.* Macrophage-released ADAMTS1 promotes muscle stem cell activation. *Nat. Commun.* **8**, 1–11 (2017).
15. Rodgers, J. T. *et al.* MTORC1 controls the adaptive transition of quiescent stem cells from G0 to GAlert. *Nature* **510**, 393–396 (2014).
16. Lindqvist, J. *et al.* Nestin contributes to skeletal muscle homeostasis and regeneration. *J. Cell Sci.* **130**, 2833–2842 (2017).

17. Armand, A. S. *et al.* Apoptosis-inducing factor regulates skeletal muscle progenitor cell number and muscle phenotype. *PLoS One* **6**, 27283 (2011).
18. White, R. B., Biérinx, A. S., Gnocchi, V. F. & Zammit, P. S. Dynamics of muscle fibre growth during postnatal mouse development. *BMC Dev. Biol.* **10**, 1–11 (2010).
19. Goel, A. J., Rieder, M. K., Arnold, H. H., Radice, G. L. & Krauss, R. S. Niche Cadherins Control the Quiescence-to-Activation Transition in Muscle Stem Cells. *Cell Rep.* **21**, 2236–2250 (2017).
20. Kuang, S., Kuroda, K., Le Grand, F. & Rudnicki, M. A. Asymmetric Self-Renewal and Commitment of Satellite Stem Cells in Muscle. *Cell* **129**, 999–1010 (2007).
21. de Jonge, W. J., Brok, M., Kemmeren, P. & Holstege, F. C. P. An Optimized Chromatin Immunoprecipitation Protocol for Quantification of Protein-DNA Interactions. *STAR Protoc.* **1**, 100020 (2020).
22. Tosic, M. *et al.* Lsd1 regulates skeletal muscle regeneration and directs the fate of satellite cells. *Nat. Commun.* **9**, 1–14 (2018).
23. Philippos, M. *et al.* A critical requirement for notch signaling in maintenance of the quiescent skeletal muscle stem cell state. *Stem Cells* **30**, 243–252 (2012).
24. Alonso-Martin, S. *et al.* Sox factors regulate murine satellite cell self-renewal and function through inhibition of β -catenin activity. *Elife* **7**, (2018).
25. Bae, J. H. *et al.* Satellite cell-specific ablation of Cdon impairs integrin activation, FGF signalling, and muscle regeneration. *J. Cachexia. Sarcopenia Muscle* **11**, 1089–1103 (2020).
26. Yamamoto, M. *et al.* Loss of MyoD and Myf5 in Skeletal Muscle Stem Cells Results in Altered Myogenic Programming and Failed Regeneration. *Stem Cell Reports* **10**, 956–969 (2018).
27. Chen, F. *et al.* YY 1 regulates skeletal muscle regeneration through controlling metabolic reprogramming of satellite cells. *EMBO J.* **38**, (2019).
28. Bi, P. *et al.* Fusogenic micropeptide Myomixer is essential for satellite cell fusion and muscle regeneration. *Proc. Natl. Acad. Sci. U. S. A.* **115**, 3864–3869 (2018).

Reviewers' Comments:

Reviewer #2:

Remarks to the Author:

The authors have now adequately addressed all my concerns in their revised manuscript. I have no remaining comments or concerns.

Reviewer #2 (Remarks to the Author):

The authors have now adequately addressed all my concerns in their revised manuscript. I have no remaining comments or concerns.

We sincerely thank the Reviewer for her/his appreciation for our work.